# Mixtures of Neural Cellular Automata: A Stochastic Framework for Growth Modelling and Self-Organization

## Abstract

Neural Cellular Automata (NCAs) are a promising new approach to model self-organizing processes, with potential applications in life science. However, their deterministic nature limits their ability to capture the stochasticity of real-world biological and physical systems. We propose the Mixture of Neural Cellular Automata (MNCA), a novel framework incorporating the idea of mixture models into the NCA paradigm. By combining probabilistic rule assignments with intrinsic noise, MNCAs can model diverse local behaviors and reproduce the stochastic dynamics observed in biological processes.

We evaluate the effectiveness of MNCAs in three key domains: (1) synthetic simulations of tissue growth and differentiation, (2) image morphogenesis robustness, and (3) microscopy image segmentation. Results show that MNCAs achieve superior robustness to perturbations, better recapitulate real biological growth patterns, and provide interpretable rule segmentation. These findings position MNCAs as a promising tool for modeling stochastic dynamical systems and studying self-growth processes.

## 1 Introduction

Biological systems are governed by emergent properties of spatial interactions between molecules and cells, many of which have a stochastic component. Traditional modeling approaches often struggle to capture the intricate patterns emerging from these processes as they lack the necessary expressive power and scalability. Cellular Automata (CAs) have emerged decades ago as a powerful tool for simulating self-organized growth emerging from simple local rules that nonetheless produce complex global behaviors. Classical CAs are hard to parameterize and scale poorly. Neural Cellular Automata instead combine machine learning with classical CAs to learn the biological rules that give rise to the complex patterns we observe in tissues. However, standard NCAs are inherently deterministic, limiting their applicability to real-world biological systems.

Stochasticity plays a crucial role in many biological processes, such as gene expression, cellular differentiation, and carcinogenesis. Cells with identical genomes can exhibit diverse behaviors due to random fluctuations in the concentration of signaling and effector molecules. Capturing this stochastic nature is essential for accurate modeling and understanding of these phenomena. At the same time, biological rules can be arbitrarily complicated, and extracting interpretable information can be challenging.

To address these limitations, we propose the Mixture of Neural Cellular Automata (MNCA), a novel framework integrating stochasticity and clustering within the NCA paradigm. By combining mixture models with NCAs, MNCA effectively captures diverse local behaviors and inherent biological randomness, while simultaneously offering interpretable rule assignments that facilitate post-hoc analysis of the system's dynamics.

Our contributions are as follows:

- **Introduction of MNCAs:** We develop the MNCA framework that extends NCAs by incorporating stochasticity and clustering, allowing for the simulation of a more diverse set of local behaviors.

- **Robustness Analysis:** Through image morphogenesis tasks, we show that MNCAs exhibit enhanced robustness to image perturbations compared to deterministic NCAs.

- **Emergent Unsupervised Segmentation:** We demonstrate that MNCAs can autonomously segment heterogeneous cells in high-content screening images by capturing morphological features, and can steer phenotype expression through constrained rule activation.

The rest of the paper is organized as follows: Section 2 reviews the background and related work, Section 3 details the MNCA methodology, Section 4 presents experimental results, and Section 5 concludes the paper.

## 2 Background

### 2.1 Spatial Modelling of Biological Systems

Spatial mathematical modeling applied to biology helps understanding biological systems by accounting for the spatial distribution and interactions of components such as cells, molecules, or organisms. This approach is essential for capturing emergent phenomena where spatial context influences the macroscopic behavior of the system.

In biological systems, spatial organization often determines function. For instance, during embryonic development, spatial gradients of morphogens guide cell differentiation and tissue formation Rogers & Schier (2011). In self-regenerating tissue such as the intestine, stem cell identity is established through spatial interactions Beumer & Clevers (2021). Similarly, in cancer biology, the spatial distribution of cells within a tumor affects growth dynamics and responses to treatment Seferbekova et al. (2023).

Agent-based modeling is a popular choice for modeling spatially organized biological phenomena. Among the different models available stochastic cellular automata are a simple but effective approach and have been used to study a wide range of systems such as cancer growth Tari et al. (2022); Lewinsohn et al. (2023); Sottoriva et al. (2010), biological development Ermentrout & Edelstein-Keshet (1993) and ecological niches Balzter et al. (1998). Parametrization is usually obtained from experimental studies or using Approximate Bayesian Computation (ABC) Noble et al. (2022), with some recent advances trying to combine ABC with Deep Learning Cess & Finley (2023).

### 2.2 Cellular Automata

Cellular Automata (CA) are discrete computational models that simulate the physical dynamics of complex systems through simple local interactions. Historically CA where introduced by Von Neumann as a model of self-reproducing systems Von Neumann et al. (1966)

In a CA, the state of each cell $s_i \in S$ at discrete time $t$ is updated based on a local update rule:

$$s_i^{t+1} = f\left(s_i^t, \{s_j^t \mid j \in \mathcal{N}(i)\}\right), \tag{1}$$

where $\mathcal{N}(i)$ denotes the neighborhood of cell $i$, and $f$ is a deterministic function defining the update rule. This simple local interaction can lead to complex global behaviors, making CA suitable for modeling self-organization and pattern formation in biological systems.

While the update function $f$ in a standard cellular automaton is strictly deterministic, probabilistic rules are more fitting for capturing the randomness inherent in processes like biological evolution. A stochastic cellular automaton (SCA) introduces randomness into the update rule:

$$s_i^{t+1} \sim P\big(f\left(s_i^t, \{s_j^t \mid j \in \mathcal{N}(i)\}\right)\big). \tag{2}$$

Specifically, each cell's next state is drawn from a probability distribution $P$ that depends on its current state and the states of its neighbors.

### 2.3 Neural Cellular Automata

Neural Cellular Automata (NCAs) extend traditional CA by replacing the deterministic update function with a neural network Mordvintsev et al. (2020); Gilpin (2019). The update rule becomes:

$$s_i^{t+1} = s_i^t + \phi\left(s_i^t, \{s_j^t \mid j \in \mathcal{N}(i)\}; \theta_k\right), \tag{3}$$

where $\phi$ is a neural network parameterized by weights $\theta$. The neural network processes the current state of the cell and its neighbors to produce an update, which is added to the current state. This formulation allows the NCA to learn complex behaviors through back-propagation on training data, rather than relying on hand-crafted rules or poorly scalable evolutionary or sampling algorithms.

In recent years, NCAs have been expanded in several ways: Tesfaldet et al. (2022) introduced attention to the definition of state update function, Grattarola et al. (2021) extended NCAs to different topological domains such as graphs, and Menta et al. (2024) implemented NCAs as a latent space process. In the field of dynamical systems, NCAs have also been explored to approximate some classes of PDEs like Reaction-Diffusion equations Richardson et al. (2024). An approach that goes towards having more than a single rule is the work of Hernandez et al. (2021), where the authors learn an Autoencoder that maps an image to the rule that generates it. Despite their flexibility, the approaches proposed above are deterministic up to a dropout parameter that allows for asynchronous updates. This determinism practically limits their ability to model biological systems, where cell-specific and probabilistic behaviors are prevalent Noble et al. (2022).

There has indeed been some research into introducing stochasticity into NCAs, especially in the context of generative modelling. For instance, Palm et al. (2022) used NCAs as decoders within VAEs; Kalkhof et al. (2025), on the other hand, employed NCAs as denoising networks within a denoising diffusion process. Zhang et al. (2024) implemented a stochastic NCA as a hierarchical process, where a coarse-grained latent representation of the input informs the reconstruction. Zhang et al. (2021) modified the NCA to output the probability of a Bernoulli distribution.

In this work, we will mainly focus on formulations that preserve the original assumptions of cellular automata. In the context of the systems we are studying—morphogenesis, emergent group dynamics, and development—the locality assumption is essential for generating physically and biologically plausible models. Accordingly, we compare only with models like the GCA Zhang et al. (2021), in particular, in this work, we will use a GCA that learns the mean and variance of a Gaussian distribution, as it better aligns with our experimental setup.

## 3 Mixture of Neural Cellular Automata

The Mixture of Neural Cellular Automata (MNCA) is a framework that extends traditional Cellular Automata (CA) by incorporating multiple sets of local update rules within a single grid-based system. In MNCA, each cell can be governed by one of several distinct automata, allowing for the modeling of heterogeneous systems where different regions exhibit unique local interactions.

We will present and study two flavors of MCAs in this work:

- **A Mixture of Neural Cellular Automata:** where the update function for each sample in a grid is a mixture of a pool of different NCAs.

- **A Mixture of Cellular Automata with intrinsic noise:** where on top of the mixture we also allow the pool of NCAs to exhibit additional inter-cluster stochastic behaviors.

### 3.1 Mixture of Neural Cellular Automata

Consider a grid of $N$ cells, where each cell $i$ has a state $s_i^t \in S$ at time $t$ and $S$ is the set of possible states. We define a set of $K$ distinct cellular automata, each characterized by its own neural-network parametrized

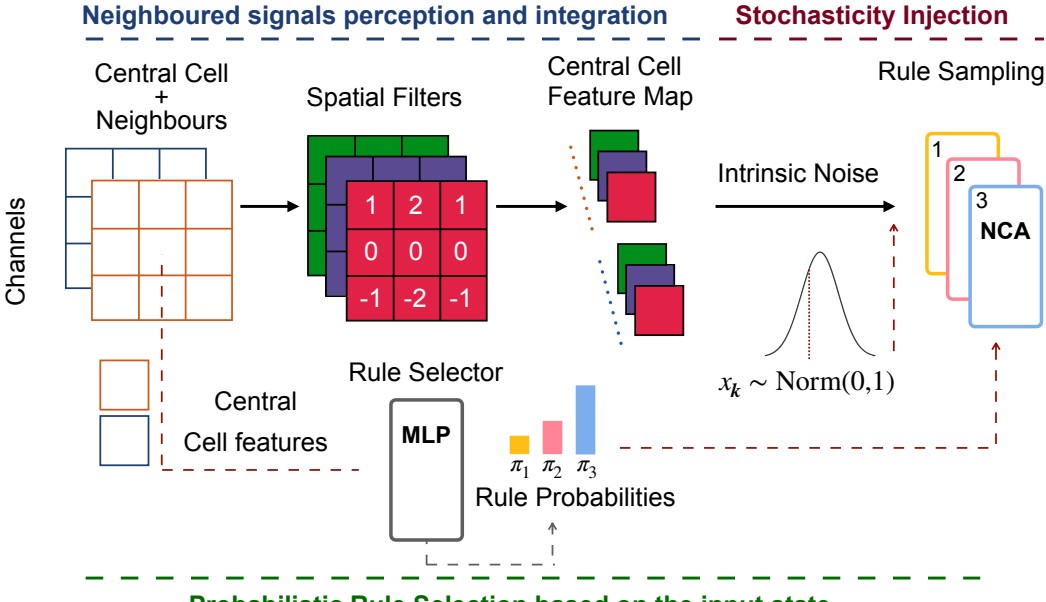

Figure 1: **Architecture of the Mixture of Neural Cellular Automata (MNCA) model with noise injection.** The model integrates signals from a central cell and its neighbours using spatial filters. The central cell's features are processed by a Multi-Layer Perceptron (MLP) that implements a Rule Selector. The Rule Selector then determines probabilities for probabilistic Neural Cellular Automata (NCA) selection. The selected NCA is fed with the feature map, and stochasticity is injected from a standard normal distribution.

transition function $\phi_k : S^{|\mathcal{N}(i)|+1} \to S$, with parameters $\theta_k$ and where $\mathcal{N}(i)$ denotes the neighborhood of cell $i$.

The state update for cell $i$ at time $t+1$ is given by:

$$z \sim \text{Cat}(\pi(s_i^t, \eta)) \tag{4a}$$

$$s_i^{t+1} = s_i^t + \prod_{k=1}^{K} \phi_k \left( s_i^t, \{s_j^t \mid j \in \mathcal{N}(i)\}; \theta_k \right)^{z_k} \tag{4b}$$

where $z_i \in \{1, 2, \ldots, K\}$ is the automaton assigned to cell $i$, $z$ is a random variable distributed according to a Categorical distribution. The Categorical is parametrized by a neural network with parameters $\eta$ and controls the rule assignment probability for each sample in the grid. To back-propagate through the Categorical distribution at training time, we use the Gumbel-Softmax trick Jang et al. (2016).

## 3.2 Mixture of Neural Cellular Automata: intrinsic noise

While the formulation above allows for different groups of local rules, we know that even well-defined biological entities are generally not constrained to behave always in a deterministic way given a particular environment. To allow more flexibility in our model we expand to incorporate a Gaussian latent vector and allow for inter-cluster stochastic updates.

$$z \sim \text{Cat}(\pi(s_i^t, \eta)) \tag{5a}$$

$$x_k \sim \text{Norm}(0, 1) \tag{5b}$$

$$s_i^{t+1} = s_i^t + \prod_{k=1}^{K} \phi_k \left( s_i^t, \{s_j^t \mid j \in \mathcal{N}(i)\}, x_k; \theta_k \right)^{z_k} \tag{5c}$$

Here, the only difference is the injected Gaussian noise $x_k$, which the network can use as an internal source of randomness to potentially implement intra-rule stochastic updates (see Appendix B for a more detailed analysis of the rationale behind this modelling choice)

## 4 Experiments

Our main motivation for studying Mixtures of Neural Cellular Automata is to explore the dynamics of cellular differentiation, carcinogenesis, and tissue morphogenesis. To show the potential of our approach, we designed a synthetic experimental setup to investigate the behavior of stem-driven growth and differentiation within a tissue.

The experiment simulates a generic epithelial tissue, incorporating five different cell types and spatial interactions among them mimicking realistic growth scenarios.

We then we studied the behavior of our model in the more classical task of image morphogenesis. Here, we show how a stochastic mixture of automata increases the stability and robustness of the pattern learned and provides an interpretable image segmentation.

Finally, we applied MNCAs to synthetic microscopy images, where they autonomously segmented cells by morphological and proteomic features, and enabled phenotype steering through constrained rule activation.

### 4.1 Model Architectures and Parameters

We maintained consistent neural network architectures across our three experiments (image morphogenesis, Visium spatial transcriptomics, and synthetic biological simulations). All models were trained using the Adam optimizer Kingma (2014) and with a Multi-Step LR Scheduling, where every time the training reaches a set of epoch milestones, the learning rate gets multiplied by a scaling factor gamma.

To keep the model simple and all rule functions $\phi_k$ follow a standard architecture with two 1 by 1 convolutional layers:

$$h = \mathrm{ReLU}(\mathrm{Conv}_{1\times 1}\mathrm{Cat}(s_i^t, \nabla_x s_i^t, \nabla_y s_i^t)) \tag{6}$$

$$s_i^{t+1} = s_i^t + \mathrm{Conv}_{1\times 1}(\mathrm{Cat}(h, x_k)) \tag{7}$$

where $s_i^t$ represents the state at time $t$, $\nabla_x$ and $\nabla_y$ are Sobel filters for spatial derivatives, $x_k$ is the gaussian noise, and Cat denotes channel-wise concatenation. This is the implementation of Equation 5c, in case of 4b and 3 the implementation is the same but with $\mathrm{Conv}_{1\times 1}(\mathrm{Cat}(h))$ instead of $\mathrm{Conv}_{1\times 1}(\mathrm{Cat}(h, x_k))$, the latter having just one of those networks.

Depending on the problem, we use the residual update in 7 or omit the $s_i^t$ sum. The network $\pi$ is implemented by a network of the same type with the only difference that the input in this case is just the current cell value $x_t$ and not $\mathrm{Cat}(x_t, \nabla_x x_t, \nabla_y x_t)$. In all experiments, we used the Mean Squared Error (MSE) against the target as our loss.

We report the parameters used in the experiment in Table 1. Other extra parameters used for the experiment in Section 4.3 are reported in Appendix D, together with an analysis of how the number of rules impacts performance.

### 4.2 Synthetic Data of Biological Development

For the configuration showed in Figure 2A, we generated a dataset of 200 realizations of simulated growth on a 35×35 grids. Each square of the grid represents a single-cell. Each realization begins with a centrally clustered

Table 1: **Model configurations across experiments.** Milestones and Gamma are the epoch and the multiplicative factor for the learning rate schedule, k stands for $10^3$

| Parameter | Tissue | Microscopy | Emoji |
|---|---|---|---|
| Channels | 6 | 24 | 16 |
| Hidden dim. | 128 | 128 | 128 |
| Rules | 5 | 5 | 6 |
| Learning rate | $1 \times 10^{-3}$ | $1 \times 10^{-3}$ | $1 \times 10^{-3}$ |
| Epochs | 800 | 8000 | 8000 |
| Residual | No | Yes | Yes |
| Dropout | 0 | 0.2 | 0.1 |
| Milestones | [500] | [5k,6k,7k] | [4k,6k,7k] |
| Gamma | 0.1 | 0.2 | 0.1 |
| Filters | $\nabla_x, \nabla_y$ | $\nabla_x, \nabla_y$ | $\nabla_x, \nabla_y$ |

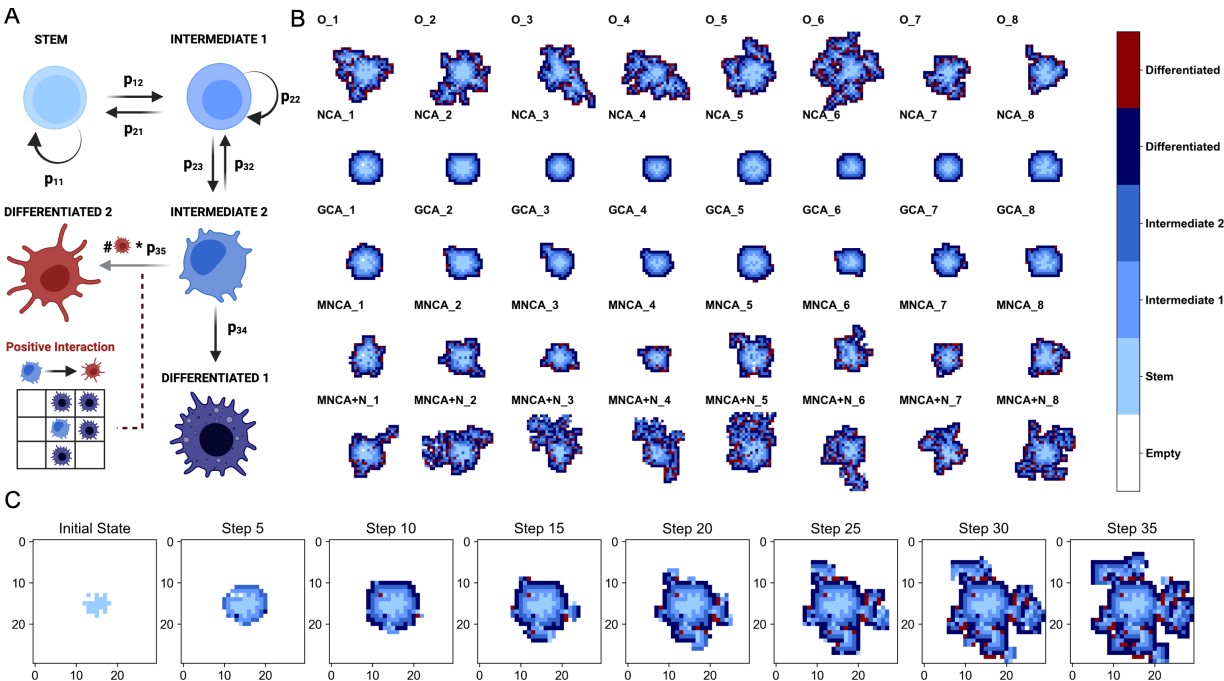

Figure 2: **Visualization of MNCA's stochastic framework applied to synthetic tissue growth.** The figure illustrates the development of cellular patterns over time, starting from an initial configuration of stem cells and evolving into differentiated tissue structures. In panel A there is a brief description of the tissue model used for the simulation. In panel B we show an example of 8 realizations of the process. In the first row we have the original final tissue, in the second row the deterministic NCA, in the third the GCA and in the last two rows the MNCA and the MNCA with intrinsic noise (O= Original, MNCA+N= MNCA with Noise). In Panel C there is an example of full tissue evolution with an MNCA starting from a random initial configuration and evolving the model for 35 steps

Table 2: **Comparison of NCA variants.** MNCA w/ N stands for MNCA with Noise. KL-div, Size-W, and Border-W are respectively the KL divergence for the cell types probability and the Wasserstein distance of the cell composition, tissue size, and border size in the generated and original dataset. For a formal definition of the metrics see Appendix C

| MODEL | KL-DIV | SIZE-W | BORDER-W |
|---|---|---|---|
| NCA | 2.057 ±0.000 | 0.547 ±0.000 | 0.430 ±0.000 |
| GNCA | 0.112 ±0.003 | 0.477 ±0.002 | 0.339 ±0.005 |
| MNCA | **0.018 ±0.001** | **0.061 ±0.008** | 0.184 ±0.010 |
| MNCA W/ INTERNAL NOISE | 0.028 ±0.001 | 0.104 ±0.012 | **0.054 ±0.013** |

group of $[5, 15]$ stem cells, evolved for 35 time steps. Stem cells have a high turnover rate and can produce either new stem cells or intermediate cells. We have two types of intermediate cells that can de-differentiate with increasingly lower probability. Eventually, those cells can divide into two types of differentiated cells: the first type is the default choice, while the second one is activated by positive interaction between cells of the first type and intermediate cells [1]. Further details on the parameter settings and the simulation procedure are provided in Appendix A.

We trained the three cellular automata introduced in Section 4.1 to reconstruct the system's state at each time step (for a formal definition of the training algorithm, see Appendix D). To provide an image-like tensor to the model we one-hot-encoded each cell-type into a single channel and gave that as input to the automata.

We then tested the consistency of reconstructed data with the training, an example of reconstruction can be found in Figure 2 B-C. Notably, the MNCAs consistently achieved higher cell-type distribution similarity than their similar-size deterministic counterpart and generated realistic tissue samples. It is particularly notable how MNCAs reduced the KL divergence of the cell type distribution of more than one order of magnitude (2.057 for the NCA against 0.018 MNCA, Table 2 ).

To quantitatively evaluate model performance (Table 2), we adopted a set of complementary metrics that capture both the statistical and spatial fidelity of the generated tissues (see Appendix C for formal definitions):

- **KL Divergence**: Measures the divergence between the cell type distribution in the real and generated data. This reflects how well the model captures the overall composition of different cell types across samples.

- **Wasserstein Distance on Tissue Size**: Compares the distribution of total occupied grid cells (i.e., tissue size) between real and generated tissues using the 1-Wasserstein distance.

- **Wasserstein Distance on Tissue Borders**: Evaluates morphological accuracy by applying a discrete Laplacian filter to the binary occupancy masks, and measuring the Wasserstein distance between the resulting border complexity scores.

These metrics jointly provide a robust quantitative framework for assessing whether the learned dynamics generate biologically realistic growth patterns that match the reference data in both cell composition and spatial structure.

Moreover, the MNCA could generate synthetic data that closely resembles real samples in terms of tissue size and spatial characteristics. At the same time, as also, quite evident from Figure 2, the deterministic NCA cannot reliably generate differentiated cells of type 2.

When then looking at the rules assignment for the MNCAs, by plotting the rule assignment probability for a given state in time, we find that the model correctly gets that approximately each cell type has a different rule. This has also been our main rationale for setting the number of rules to 5. Interestingly, the model

---

[1]In particular the rate of transition from INTERMEDIATE 2 to DIFFERENTIATED 2 is intrinsically zero. There is, however, a positive contribution fixed contribution to this rate for each DIFFERENTIATED 1 cell type in the neighborhood.

assigns a specific rule to empty space, while grouping stem and differentiated type 2 cells under a shared rule. Another rule broadly captures non-stem cells, suggesting the model distinguishes stem from non-stem behavior despite overlaps.

These results suggest that the MNCAs effectively learned the core of the underlying generative dynamics of the systems, being able to reproduce spatial patterns that closely resemble the original data distribution.

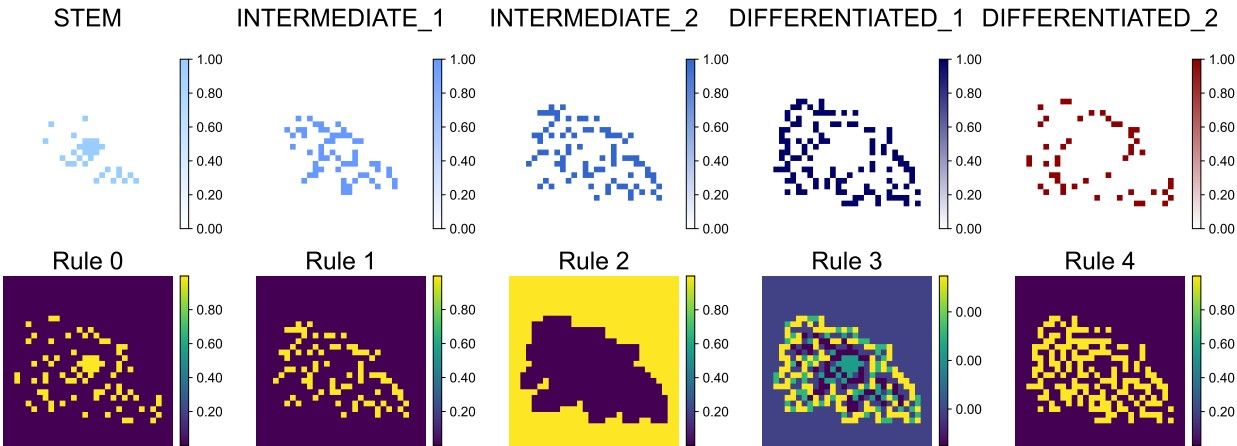

Figure 3: **Visualization of rule assignments in MNCA simulations** On the first row, we split the tissue by cell type. On the second row, we have the rules assignment probability for each rule. Different rules are approximately assigned to distinct cell types.

In contrast to the popular ABC-based ABM framework described in Section 2.1, MNCAs achieve comparable simulation quality with significantly less supervision (see Appendix E for a more comprehensive discussion). While ABC methods require substantial prior knowledge and meticulous tuning to yield satisfactory results, our black-box approach needs only a tensor-based representation of the system and does not depend on direct access to the simulation algorithm.

### 4.3   Image Morphogenesis

To assess the usefulness of our mixture-based neural cellular automata (NCA) model in a more standard computer vision context, we conducted a series of morphogenesis experiments using a subset of the Twitter emoji dataset. This has been historically the first task for which NCAs have been developed. We first train the NCA to reproduce the target image from a fixed random pixel as in Mordvintsev et al. (2020) and then perturb the image (we report the training loop in Appendix D).

We evaluated the model's ability to reconstruct the original image under three distinct perturbation scenarios:

1. **Chunk Removal:** A contiguous rectangular block of NxN pixels is removed in a random position. We tested 2 sizes of respectively 5x5px and 10x10px. To avoid empty areas we constrain the center of the box to be on the image.

2. **Gaussian Noise Addition:** Additive Gaussian noise was introduced at random on a percentage $\rho$ of pixels. We testes $\rho = [0.1, 0.25]$

3. **Sparse Pixel Removal:** Random individual pixels are randomly removed, creating scattered white areas. We test respectively 100 and 500 px.

Mixture-based NCA vastly outperforms the single-rule baseline across all perturbation scenarios. We report the results in Table 3 and plot some examples for each perturbation in Figure 4. It is interesting to note

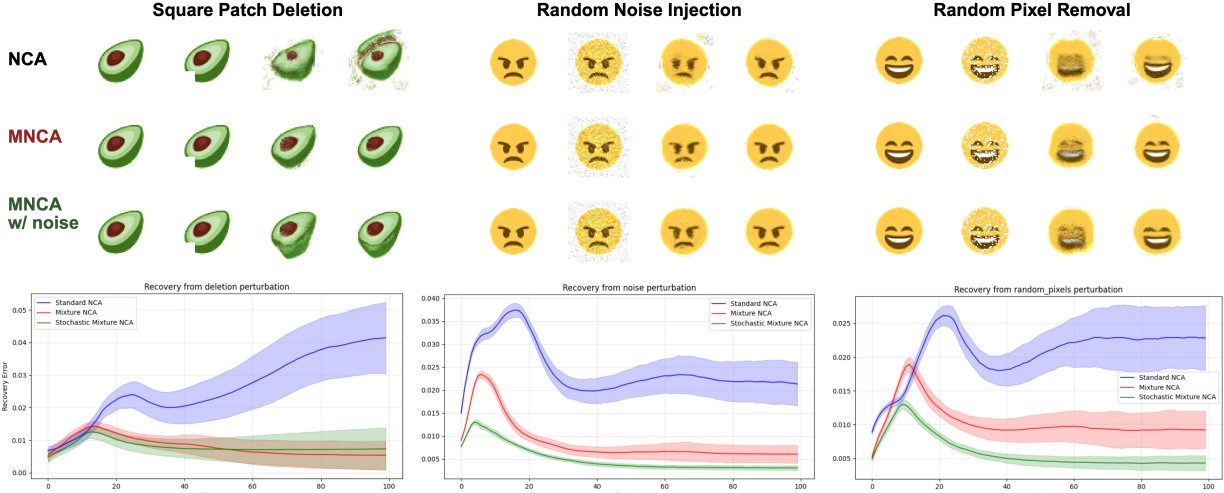

Figure 4: **MNCA in image morphogenesis.** The top half of the plot shows an example of the 3 perturbation types we studied in the paper. The columns are respectively, the original image, the perturbed image, the image after 50 steps of recovery, and the final recovered image after 100 steps. Each row shows a separate model. The bottom half of the plot shows the MSE error over time. We see how the NCA can diverge, while the MNCAs, after an initial peak in the error, go back to the original image. Confidence intervals (95%) were computed across 50 different perturbations of the same kind. We do not show the GCA as it shows no clear benefit over the NCA.

how none MNCAs the two different flavors of MNCA, while still outcompeting the deterministic NCA, have the best performance across all images. On the other end, the GCA seem not to have a major advantage over the vanilla NCA in terms of robustness. We believe that this could be due to more difficulties in the convergence of the model in Equation 2. Another observation regards the nature of the perturbation; it looks like a localized perturbation, like square patch deletion, tends to have lower error, while global noise addition has a way worse effect.

When we look at the reconstruction error over time, we noted (reported in Figure 4) that the models initially tend to overshoot and depart a lot from the original image and, after a peak they come back to their original image attractor.

We believe it is important to note that in Mordvintsev et al. (2020) the authors train an NCA that is resistant to perturbation by applying the perturbation during training. On the contrary in this experiment, any tolerance to the perturbation is a purely emergent phenomenon.

We can look again at the rule assignment on the images. Notably, the model seems to be able to segment relevant parts of the image. In particular in Figure 5 the model uses different rules for the outside white pixels, for the contours of the emoji, and for some internal details like the eyes.

## 4.4 Microscopy Images Classification

While Neural Cellular Automata (NCAs) have primarily been used for image and pattern generation, a seminal study demonstrated their potential for supervised digit classification on the MNIST dataset Randazzo et al. (2020), addressing whether agents following identical rules can develop a communication protocol to infer their assigned digit through repeated interactions.

Building on our model's capacity to segment inputs in an interpretable manner and combine rule mixtures, we explored its ability to classify and structure objects, this time, however, in a fully unsupervised setting.

We used image set BBBC031v1 Piccinini et al. (2017), available from the Broad Bioimage Benchmark Collection Ljosa et al. (2012), which contains synthetic high-content screening (HCS) data simulating drug-

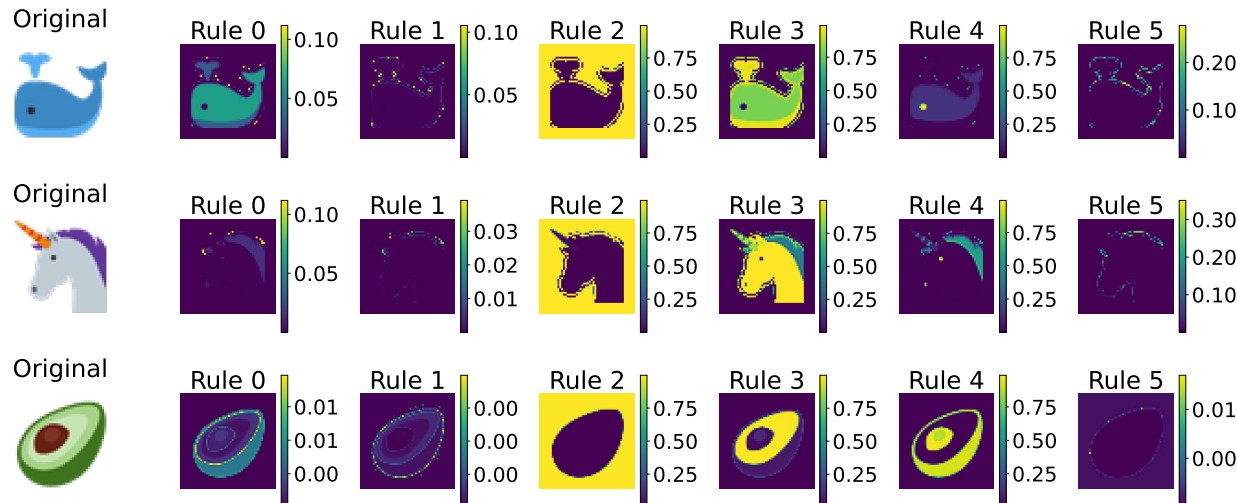

Figure 5: **Visualization of rule assignments in MNCA on emojis.** Different rules are assigned to distinct parts of the image. In particular, the body of the image, the empty space, and the borders tend to have specific rules.

induced perturbations of cell shape and protein expression (represented by different colors). We used the experimental setup from Section 4.3 and initialized the seeds as the centroids of each cell (Figure 6A)

First, we evaluated whether the MNCA could internally differentiate and assign distinct rule sets to different cell types purely based on local observations and interactions. The resulting rule distributions, visualized in Figure 6B, revealed that the model naturally grouped cells according to their morphological and proteomic profiles, effectively segmenting the tissue into interpretable subtypes—without the need for supervision.

The ground truth dataset provides a continuous morphological feature known as the Cell Shape Parameter, which quantifies the irregularity of each synthetic cell's contour. As illustrated in Figure 6C, the model's inferred rule assignments show a clear correlation with this parameter, indicating that the MNCA captures intrinsic shape-based features during development.

We report here the main intuition behind this value (see Lehmussola et al. (2007) for a more in depth discussion). Cells are simulated using a polygonal model, where the coordinates of each vertex are given by:

$$
\begin{aligned}
x_i &= r\left[U(-\alpha, \alpha) + \cos\left(\theta_i + U(-\beta, \beta)\right)\right] \\
y_i &= r\left[U(-\alpha, \alpha) + \sin\left(\theta_i + U(-\beta, \beta)\right)\right]
\end{aligned}
\tag{8}
$$

Here, $r$ is the base radius, $\theta_i$ is the angular position of the $i$-th vertex (evenly spaced in $[0, 2\pi]$), and $U(a, b)$ denotes a uniform random variable in the interval $[a, b]$. The parameters $\alpha$ and $\beta$ respectively control the radial and angular distortions of the cell shape. The Cell Shape Parameter is the value of $\alpha$ and $\beta$ and correlates with the degree of morphological deformation: higher values produce more irregular, jagged shapes, while lower values yield smoother, rounded contours.

Finally, we explored whether the MNCA could be directed toward specific phenotypes by constraining it to use only specific class-defining rule during inference. This manipulation effectively steered the entire cellular population toward different target phenotypes, showcasing the model's potential not only for passive classification but also for active control over emergent morphologies (Figure 7).

These findings indicate that MNCAs are capable of capturing structural differences in spatially organized cell populations, suggesting potential use for modeling simple biological processes through local rule-based dynamics.

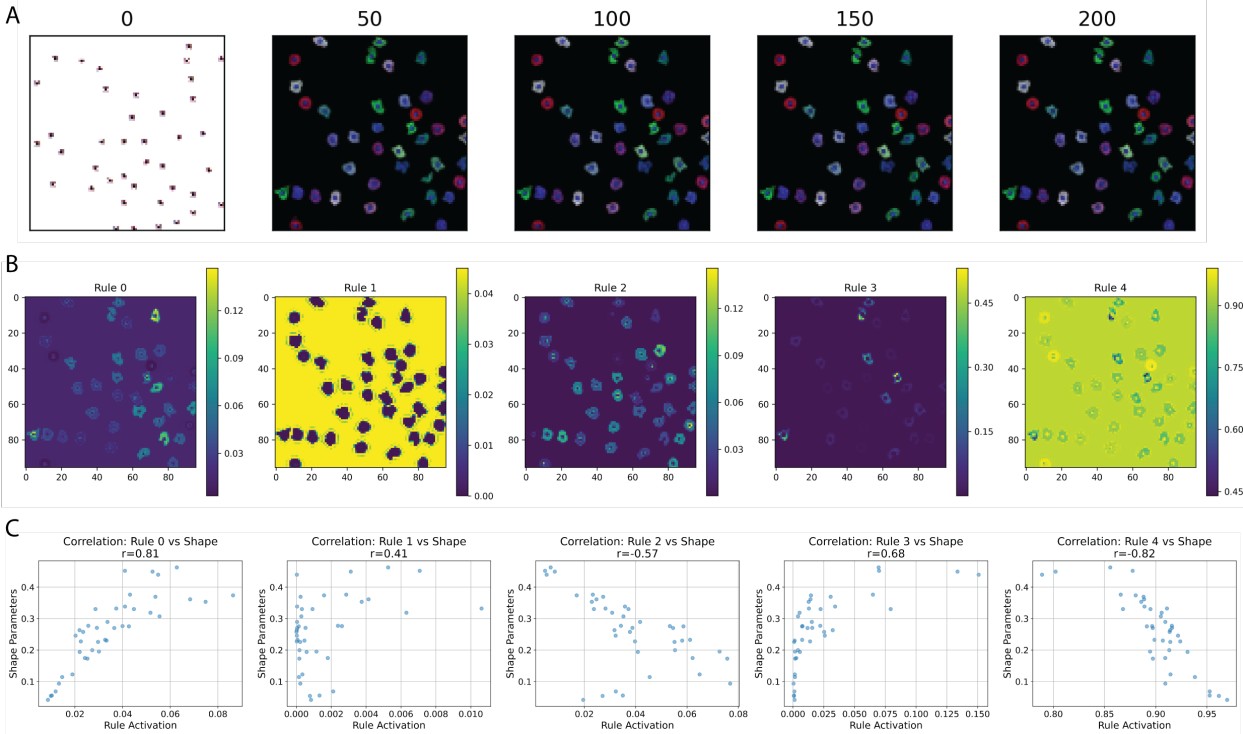

Figure 6: **MNCA fitting to cell microscopy images.** A: Morphogenesis process initialized from fixed seeds, with one seed placed at each cell centroid in the input image. B: Probabilistic rule assignments for each cell, as inferred by the model. C: Correlation analysis between rule assignments and a cell shape parameter, where higher values indicate more irregular morphologies.

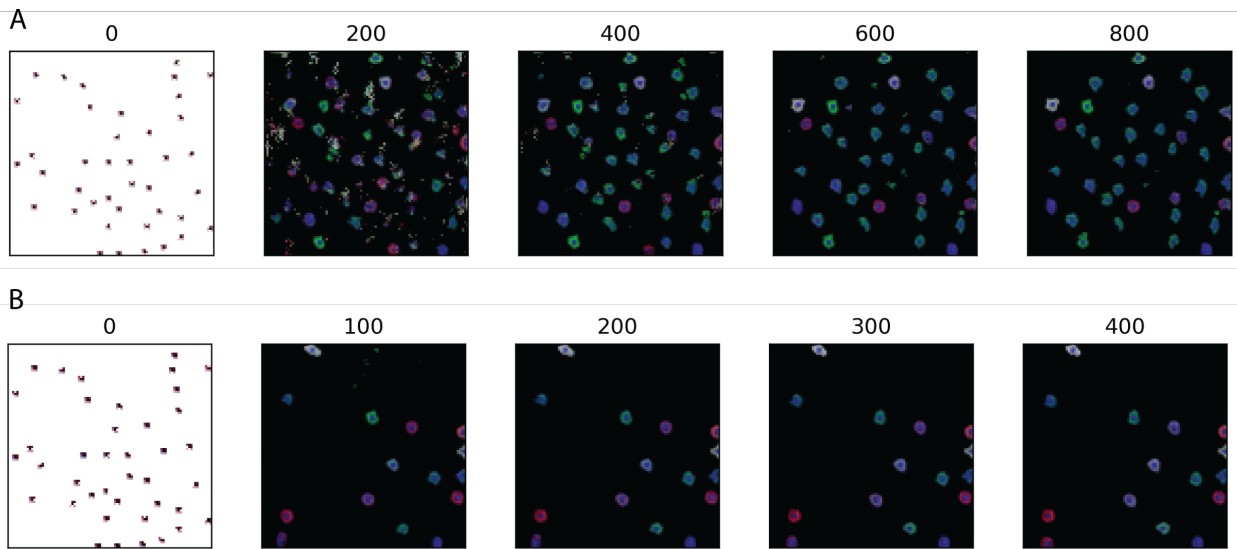

Figure 7: **Steering the evolution of the system by changing the rule probabilities.** A: Morphogenesis after reducing 100 times the probability of rule 0. The new image shows smaller and rounder cells. B: Morphogenesis after doubling the probability of the rule 5. Most of the irregular cells are not able to fully develop and disappear.

Table 3: **Performance Across Different Perturbations on a set of 6 emojis**.Here the metric is the Mean Squared Error (MSE) after 100 steps of recovery and on a batch of 50 different perturbations of each kind

| Model | Del 5x5px | Del 10x10px | Noise 10% | Noise 25% | Removal 100px | Removal 500px |
|---|---|---|---|---|---|---|
| 🫧 | | | | | | |
| NCA | 0.024 ±0.008 | 0.027 ±0.009 | 0.049 ±0.007 | 0.052 ±0.010 | 0.026 ±0.006 | 0.032 ±0.011 |
| GCA | 0.017 ±0.005 | 0.019 ±0.006 | 0.022 ±0.005 | 0.025 ±0.005 | 0.016 ±0.004 | 0.021 ±0.004 |
| MNCA | **0.007 ±0.003** | **0.008 ±0.003** | 0.071 ±0.002 | 0.079 ±0.004 | **0.007 ±0.003** | **0.010 ±0.003** |
| MNCA w/ Noise | 0.008 ±0.002 | 0.010 ±0.003 | **0.009 ±0.004** | **0.013 ±0.006** | 0.009 ±0.002 | 0.011 ±0.003 |
| 😄 | | | | | | |
| NCA | 0.025 ±0.005 | 0.027 ±0.006 | 0.025 ±0.007 | 0.030 ±0.006 | 0.025 ±0.005 | 0.030 ±0.006 |
| GCA | 0.018 ±0.007 | 0.024 ±0.010 | 0.022 ±0.007 | 0.024 ±0.006 | 0.019 ±0.007 | 0.031 ±0.008 |
| MNCA | 0.016 ±0.008 | 0.026 ±0.009 | 0.013 ±0.006 | **0.018 ±0.005** | 0.014 ±0.005 | **0.021 ±0.005** |
| MNCA w/ Noise | **0.007 ±0.003** | **0.016 ±0.006** | **0.008 ±0.002** | 0.026 ±0.005 | **0.008 ±0.003** | 0.024 ±0.005 |
| 😠 | | | | | | |
| NCA | 0.018 ±0.004 | 0.021 ±0.006 | 0.022 ±0.006 | 0.026 ±0.007 | 0.019 ±0.006 | 0.025 ±0.007 |
| GCA | 0.018 ±0.005 | 0.023 ±0.008 | 0.019 ±0.005 | 0.025 ±0.004 | 0.020 ±0.006 | 0.028 ±0.008 |
| MNCA | **0.009 ±0.002** | 0.017 ±0.005 | 0.011 ±0.002 | **0.012 ±0.003** | 0.010 ±0.002 | 0.015 ±0.003 |
| MNCA w/ Noise | **0.009 ±0.003** | **0.015 ±0.004** | **0.008 ±0.002** | 0.016 ±0.003 | **0.009 ±0.003** | **0.013 ±0.003** |
| 🤔 | | | | | | |
| NCA | 0.014 ±0.003 | 0.018 ±0.005 | 0.014 ±0.003 | 0.017 ±0.004 | 0.014 ±0.004 | 0.022 ±0.004 |
| GCA | 0.016 ±0.004 | 0.021 ±0.006 | 0.021 ±0.005 | 0.024 ±0.006 | 0.018 ±0.005 | 0.023 ±0.006 |
| MNCA | 0.009 ±0.004 | 0.021 ±0.008 | 0.009 ±0.003 | 0.013 ±0.005 | 0.008 ±0.003 | 0.019 ±0.007 |
| MNCA w/ Noise | **0.006 ±0.002** | **0.015 ±0.006** | **0.005 ±0.001** | **0.009 ±0.002** | **0.007 ±0.002** | **0.018 ±0.003** |
| 🥑 | | | | | | |
| NCA | 0.035 ±0.018 | 0.056 ±0.026 | 0.073 ±0.013 | 0.101 ±0.021 | 0.032 ±0.018 | 0.097 ±0.028 |
| GCA | 0.012 ±0.006 | 0.026 ±0.014 | 0.044 ±0.007 | 0.046 ±0.015 | 0.011 ±0.004 | 0.035 ±0.008 |
| MNCA | **0.002 ±0.001** | **0.006 ±0.004** | **0.003 ±0.001** | **0.008 ±0.005** | **0.003 ±0.001** | **0.006 ±0.001** |
| MNCA w/ Noise | 0.006 ±0.002 | 0.011 ±0.004 | 0.005 ±0.002 | 0.009 ±0.002 | 0.006 ±0.002 | 0.010 ±0.003 |
| 🦎 | | | | | | |
| NCA | 0.021 ±0.008 | 0.030 ±0.016 | 0.080 ±0.010 | 0.079 ±0.028 | 0.023 ±0.006 | 0.032 ±0.009 |
| GCA | 0.023 ±0.009 | 0.032 ±0.014 | 0.079 ±0.010 | 0.083 ±0.013 | 0.026 ±0.007 | 0.038 ±0.012 |
| MNCA | **0.009 ±0.002** | **0.013 ±0.005** | **0.009 ±0.002** | **0.012 ±0.002** | **0.009 ±0.002** | **0.012 ±0.003** |
| MNCA w/ Noise | 0.022 ±0.008 | 0.029 ±0.012 | 0.025 ±0.009 | 0.039 ±0.012 | 0.023 ±0.010 | 0.030 ±0.012 |

## 5 Conclusion

In this work, we extended the concept of Neural Cellular Automata (NCA) by introducing stochasticity through a mixture of rules, hence proposing the Mixture of Neural Cellular Automata (MNCA).

Using a combination of synthetic experiments and analyses of real spatial transcriptomic data, we demonstrated that MNCAs can effectively simulate complex biological systems. These results highlight the potential of leveraging high-throughput spatial data or computationally expensive simulation pipelines to train cellular automata models, even in scenarios where the parameter space is highly complex or the underlying rules are uncertain: conditions that typically challenge standard SCA learning methods.

Moreover, we showed that MNCAs not only excel in modeling dynamic systems but also exhibit enhanced robustness to a wide range of perturbations compared to deterministic NCAs in the context of image morphogenesis. An additional benefit of this framework is its interpretability: analyzing the rule assignments provides insights into the learned behavior of the model.

This work paves the way for numerous exciting future directions. Upcoming research will focus on scaling MNCAs to larger systems and networks, improving the interpretability of the learned rules, and enhancing the model's ability to handle incomplete time-series data. Additionally, the general framework presented here can be tailored to the unique structures and challenges of the different spatial biology technologies currently available.

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

# A  Synthetic Simulation of Tissue Growth

This model simulates the development and maintenance of tissue organization from an initial cluster of stem cells. It captures key biological features of stem cell-driven tissue organization, which is particularly relevant for studying systems such as intestinal crypts or clonal hematopoiesis.

The model incorporates several fundamental biological principles:

- **Hierarchical Cell Organization:** The tissue is organized in a hierarchy of cell types, from stem cells through intermediate progenitors to fully differentiated cells, reflecting the organization observed in many epithelial tissues.

- **Local Cell Interactions:** Cell fate decisions are influenced by the local cellular environment, mimicking the role of signaling niches in tissue organization.

- **Differential Division Rates:** Different cell types exhibit distinct proliferation rates, with stem cells and early progenitors showing higher division rates compared to differentiated cells.

- **Cell Type-Specific Survival:** The model implements different death and growth rates for each cell type, reflecting the biological reality where stem cells are more protected while differentiated cells undergo regular turnover.

The model implements a stochastic process where each cell is synchronously updated at each time step based on a set of kinetic parameters. A full description of the procedure is in Algorithm 1

Here we report the parameters we used in our experiment:

The system comprises five distinct cell types $\mathcal{T} = \{\text{STEM}, \text{INT1}, \text{INT2}, \text{DIFF1}, \text{DIFF2}\}$, with dynamics governed by division, death, and survival rates.

The division rates $b$ are defined as: $b_{\text{stem}} = 0.8$, $b_{\text{int1}} = 0.5$, $b_{\text{int2}} = 0.5$, for differentiated cells $b$ is set to 0. Death rates $d$ are specified as: $d_{\text{stem}} = 0$, $d_{\text{int1}} = 0$, $d_{\text{int2}} = 0$, $d_{\text{diff1}} = 0.001$, $d_{\text{diff2}} = 0.001$. Survival rates $s$ follow: $s_{\text{stem}} = 0$, $s_{\text{int1}} = 0$, $s{\text{int2}} = 0.01$, $s_{\text{diff1}} = 1.0$, $s_{\text{diff2}} = 1.0$.

The base differentiation probabilities are defined by matrix $\mathbf{D} \in \mathbb{R}^{5 \times 5}$:

$$\mathbf{D} = \begin{pmatrix} 0.3 & 0.8 & 0.0 & 0.0 & 0.0 \\ 0.1 & 0.2 & 0.8 & 0.0 & 0.0 \\ 0.0 & 0.0 & 0.2 & 1.0 & 0.0 \\ 0.0 & 0.0 & 0.0 & 1.0 & 0.0 \\ 0.0 & 0.0 & 0.0 & 0.0 & 1.0 \end{pmatrix} \tag{9}$$

where $D_{ij}$ represents the rate of transitioning from type $i$ to type $j$.

Cell-cell interactions are modeled through the interaction matrix $\mathbf{I} \in \mathbb{R}^{5 \times 5}$:

$$\mathbf{I} = \begin{pmatrix} 0.0 & 0.0 & 0.0 & 0.0 & 0.0 \\ 0.0 & 0.0 & 0.0 & 0.0 & 0.0 \\ 0.0 & 0.0 & 0.0 & 0.0 & 0.0 \\ 0.0 & 0.0 & 0.0 & 0.0 & 0.3 \\ 0.0 & 0.0 & 0.0 & 0.0 & 0.0 \end{pmatrix} \tag{10}$$

where $I_{ij}$ modifies the differentiation rate based on neighboring cells of type $j$ for cells of type $i$.

At each time step $t$, cells undergo stochastic events (division, death, or survival) with probabilities normalized by the total rate. For instance, for the cell type *stem* the probability of dying would be: $P(\text{event}) = d_{stem}/(b_{stem} + d_{stem} + s_{stem})$. During division events, daughter cells may differentiate according to the probabilities in $\mathbf{D}$, modified by neighboring cells through $\mathbf{I}$. The division is allowed only on empty cells in the Moore neighbor; otherwise, the cell survives.

The model's simplicity and incorporation of key biological principles make it a useful tool for understanding. While it necessarily abstracts many biological details, it captures essential features that drive tissue organization and maintenance.

---

**Algorithm 1** Tissue Growth Simulation

---

    **Input:** grid size $N$, initial stem cells $n_s$
    **Input:** cell rates $\mathbf{R} = \{b_\tau, d_\tau, s_\tau\}$ for each cell type $\tau \in \mathcal{T}$, transition matrices $\{\mathbf{D}, \mathbf{I}\}$
    Initialize $G_0 \in \mathbb{Z}^{N \times N}$ with $n_s$ stem cells at random positions
    **repeat**
      **for** each cell $c$ at position $(x, y)$ in $G_t$ **do**
        **if** $c \neq$ EMPTY **then**
          $\rho \sim \mathcal{U}(0, 1)$ {Sample uniform random variable}
          $\mathbf{p} = \mathbf{R}(c)/\|\mathbf{R}(c)\|_1$ {Normalize rates to probabilities}
          **if** $\rho < p_{death}$ **then**
            $G_{t+1}(x, y) \leftarrow$ EMPTY
          **else if** $\rho < p_{death} + p_{div}$ **then**
            $\mathcal{N} \leftarrow$ EmptyNeighbors$(x, y)$
            **if** $\mathcal{N} \neq \emptyset$ **then**
              Sample $(i, j) \sim$ Uniform$(\mathcal{N})$ {Pick a Random Empty Neighbour}
              $\mathbf{k} = \mathbf{B}(c) + \sum_{n \in \mathcal{N}} \mathbf{I}(n)$ {Generate a vector of rates for cell-type division}
              $\mathbf{k} = \mathbf{k}/\|\mathbf{k}\|_1$ {Normalize to probabilities}
              $G_{t+1}(i, j) \sim$ Categorical$(\mathbf{k})$
            **end if**
          **end if**
        **end if**
      **end for**
      $t \leftarrow t + 1$
    **until** $t = T$

---

## B   Effect of Internal Stochasticity in MNCA

This appendix investigates the impact of internal stochastic noise $x_t$ on the flexibility and representational power of Mixture of Neural Cellular Automata (MNCA).

While an optimal set of MNCA rules might, in principle, explicitly represent all stochastic state transitions, determining this set is generally impractical due to unknown or highly sparse transitions. Consequently, rare events or unknown stochastic events could remain unmodeled even in the mixture setup.

To evaluate the practical benefits of internal stochasticity, we constructed a minimalistic simulation involving two cell types to separate the contribution of the internal noise and the rule sampling component. We used stem cells and differentiated cells. The experimental conditions included a slightly elevated stochastic cell death rate, generating infrequent, sparse cell-death events that manifest as localized empty spaces (see Figure 8 first row).

In particular, we used division rates $b_{\text{stem}} = 0.8$ and $b_{\text{diff}} = 0$. Death rates $d_{\text{stem}} = 0.05$ and $d_{\text{diff}} = 0$. Survival rates $s$ $s_{\text{stem}} = 1$ and $s_{\text{diff}} = 1$ The differentiation probability matrix was:

$$\mathbf{D} = \begin{pmatrix} 0.0 & 0.0 & 0.0 & 0.1 & 0.0 \\ 0.0 & 0.0 & 0.0 & 0.0 & 0.0 \\ 0.0 & 0.0 & 0.0 & 0.0 & 0.0 \\ 0.0 & 0.0 & 0.0 & 1.0 & 0.0 \\ 0.0 & 0.0 & 0.0 & 0.0 & 0.0 \end{pmatrix} \tag{11}$$

The interaction matrix was all set to zero.

In this controlled scenario, a deterministic MNCA with a biologically informed heuristic (one rule per cell type) failed to adequately capture rare stochastic dynamics. Conversely, an MNCA endowed with internal Gaussian noise successfully leveraged portions of this stochasticity to approximate these rare phenomena.

Comparative training results (Figure 8) confirmed the noise-enabled MNCA's ability to reproduce a higher rate of stochastic cell-death patterns.

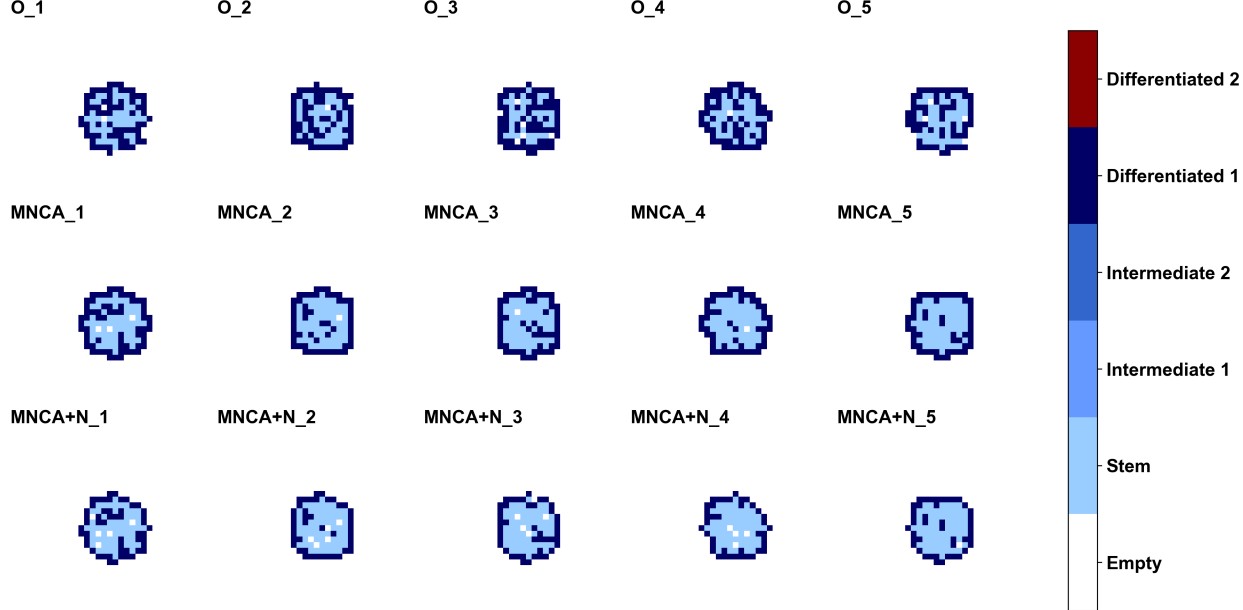

Figure 8: **Comparative visualization of the vanilla MNCA versus the internal noise version.** The top row is are trajectory from the training dataset. The middle row is the vanilla MNCA, and the last row is the MNCA with internal noise $x_t$. Notably, the last row shows a higher percentage of stochastic death events.

To rigorously quantify this phenomenon, we conducted an additional controlled experiment. We employed deterministic rule selection (consistently selecting the highest probability rule) but retained internal Gaussian noise. Analyzing 1000 stochastic updates initialized identically, we examined the correlation between noise values and subsequent cell-state predictions.

Results showed clear partitioning of the Gaussian noise distribution, with rare events ( 4-5%) consistently driven by the distribution tails. Remarkably, the MNCA implicitly represented these events both through the rare but consistent emergence of empty cells. (Figure 9).

These results underline the essential role of internal stochasticity in MNCA, particularly in modeling complex, real-world biological processes characterized by rare and unpredictable transitions.

## C   Evaluation Metrics for Neural Cellular Automata Models

We evaluate our Neural Cellular Automata (NCA) models using different complementary metrics that capture complementary aspects of the generated cellular patterns. Since our models aim to generate realistic tissue patterns with multiple cell types, we need metrics that assess both the statistical distribution of cell types and their spatial organization.

To compare the real and generated tissues, we employ three complementary measures:

- **Kullback-Leibler (KL) Divergence on cell type proportions:** $D_{KL}(P|Q) = \sum_i P(i) \log \frac{P(i)}{Q(i)}$ Where $P$ is the true cell type distribution in the whole cohort and $Q$ is the cell type distribution in

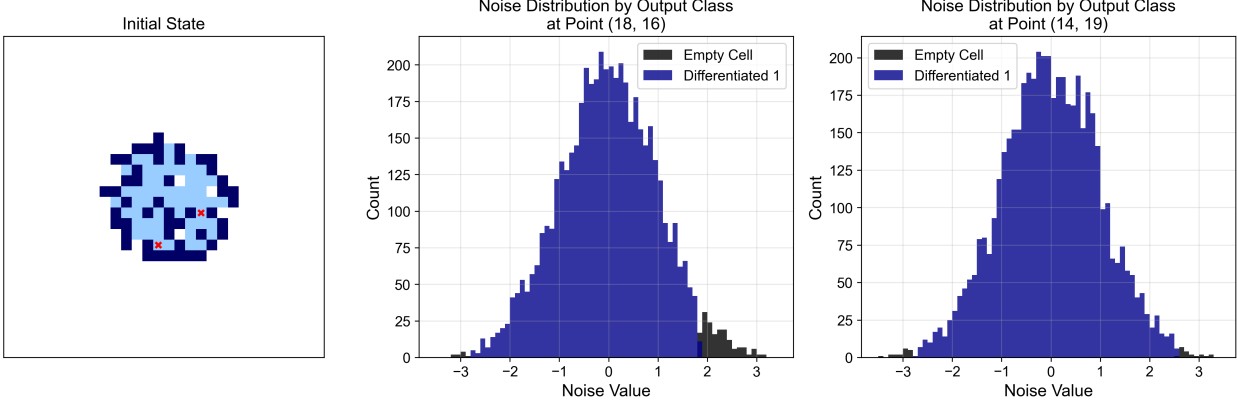

Figure 9: **Visualization of internal Gaussian noise distributions driving cell-state predictions in MNCA.** The left panel shows the initial cell-state configuration, highlighting two specific spatial points. The center and right panels illustrate the noise-value distributions partitioned by output class at the indicated coordinates. These plots demonstrate how MNCA leverages noise to implicitly model state transitions by exploiting the tails of the distribution.

the generated sample. This metric is useful for measuring how well our NCA models capture the correct proportions of different cell types.

- **Wasserstein Distance of Tissue Size Distribution:** For each real and generated data tissue we compute its size as simply the sum of all non-empty spots. We then compare the size distribution in real $U$ and generated $V$ using the 1-Wasserstein distance amongst empirical distributions, which for 1D distributions is simply:

$$W_1(U, V) = \int_{-\infty}^{\infty} |F_U(x) - F_V(x)| \, dx \tag{12}$$

Where $F$ is the empirical Cumulative Distribution Functions $F = \frac{1}{n} \sum_{i=1}^{n} \mathbf{1} \, (x_i \leq x)$

- **Wasserstein Distance of Tissue Border Size Distribution:** For each real and generated data tissue we compute a border size metric as:

$$B = \sum_{i,j} |(\nabla^2 M)_{i,j}| > \theta \tag{13}$$

where $M$ is the binary mask of cell occupancy (1 for cells, 0 for empty space), $\nabla^2$ is the discrete Laplacian operator implemented as a $3 \times 3$ convolution kernel and $\theta = 0.1$ is a threshold parameter:

$$K = \frac{1}{8} \begin{bmatrix} -1 & -1 & -1 \\ -1 & 8 & -1 \\ -1 & -1 & -1 \end{bmatrix} \tag{14}$$

The combination of these metrics allows us to compare different NCA architectures and evaluate their ability to capture both the statistical and structural properties of real biological tissues. This is essential for developing NCA models that can not only match cell type distributions but also generate spatially coherent and biologically plausible tissue patterns.

We then again compute the 1-Wasserstein distance between the border statistics of real and generated data.

# D   Training routines for the Neural Cellular Automata and Extra Parameters

We use a simple algorithm (described in Algorithm 2) to train our automata on the biological time series data. For each epoch with a specific time-window size, the algorithm samples a random part of the time series and learns to reconstruct it by computing the loss with the original realization every $\tau$ steps. The $\tau$ parameter is always one, as the probabilities are constant over time. We believe this is the best method to extract all possible information from the time series.

---

**Algorithm 2** NCA Training for Biological Time-Series

---

1: **Input:** target sequences $\{S_1, ..., S_n\}$,sequence length $T$
2: **Input:** window size $w$, number of cell types $K$, epochs $E$,
3: **Input:** evolution steps of the automata $\tau$, number of tissue samples $M$, small stability constant $\epsilon$
4: Initialize model parameters $\theta$ randomly
5: Initialize optimizer
6: **for** epoch = 1 **to** $E$ **do**
7:    $t_{start} \sim \mathcal{U}(0, T - w)$ {Sample random window}
8:    **for** $t = t_{start}$ **to** $t_{start} + w$ **step** $\tau$ **do**
9:       $X_t \leftarrow S_t$ {Encode states}
10:       $Y_t \leftarrow S_{t+\tau}$ {Future states}
11:       **for** $t_{pred} = t$ **to** $t + \tau$ **step** 1 **do**
12:          **if** $t_{pred} == t$ **then**
13:             $\hat{Y}_t \leftarrow f_\theta(X_t)$ {NCA prediction, from input}
14:          **else**
15:             $\hat{Y}_t \leftarrow f_\theta(\hat{Y}_t)$ {NCA prediction, from evolved input}
16:          **end if**
17:       **end for**
18:       // Compute loss and update
19:       $\mathcal{L} \leftarrow \frac{1}{M}\text{MSE}(Y_t, \hat{Y}_t)$
20:       $\nabla\theta \leftarrow \nabla_\theta \mathcal{L}$
21:       $\nabla\theta \leftarrow \frac{\nabla\theta}{||\nabla\theta||+\epsilon}$ {Normalize gradients}
22:       Update $\theta$ using optimizer
23:    **end for**
24: **end for**

---

The training algorithm for Neural Cellular Automata (NCA) introduced in Mordvintsev et al. (2020) is actually more complicated then the one we used above, mainly because in the task of image morphogenesis the model has to evolve without supervision for a long time. They introduced a pool-based training strategy that combines gradient descent with principles from evolutionary algorithms. By maintaining a pool of growing patterns and selectively replacing the worst-performing samples with seed states, this approach helps prevent pattern collapse and promotes the discovery of robust growth trajectories. We used this training strategy in our experiment in Section 4.3 and 4.4, we report the algorithm here to help the readers.

For our experiment on emojis, we used a batch size $B$ of 8, a state dimension $D$ of 16 (4 RGBA channels + 12 internal extra states initialized to 0), and a pool size $P$ of 1000 automata. Emojis are resized to a 40x40px image and padded with 6 white pixels on all four side. As seed location we used the pixel $[30, 50]$, $n_{min}$, and $n_{max}$ were set to 30 and 50, respectively.

For the microscopy experiment, we used again a batch size $B$ of 8, a state dimension $D$ of 24 (4 RGBA channels + 20 internal extra states initialized to 0), and a pool size $P$ of 600 automata. Images are resized to 96x96px without padding. As seed location we used the centroid pixel of each cell, $n_{min}$ and $n_{max}$ were set to 30 and 50, respectively.

All experiments were run on a single NVIDIA Tesla V100-SXM2-32GB.

To evaluate the impact of rule complexity, we analyze the KL divergence between the simulated and target cell-type distributions as a function of the number of mixture rules. As illustrated in Figure 10, models with

---

**Algorithm 3** Pool-Based NCA Training from Mordvintsev et al. (2020)

---
1: **Input:** Target image $I$, pool size $P$, batch size $B$, state dimension $D$, number of steps $T$
2: **Input:** Small stability constant $\epsilon$, seed location $(s_h, s_w)$, growth steps range $n_{min}, n_{max}$
3: Initialize model parameters $\theta$ randomly
4: Initialize optimizer
5: $x_0 \leftarrow \mathbf{0}^{B \times D \times H \times W}$ {Initial state}
6: $x_0[..., 3 :, s_h, s_w] \leftarrow 1$ {Place seed at $(s_h, s_w)$}
7: pool $\leftarrow \{x_0\}^P$ {Initialize pool with P copies}
8: **for** step $= 1$ **to** total_steps **do**
9: $\quad \mathcal{B} \leftarrow$ SampleBatch(pool, $B$) {Sample batch from pool}
10: $\quad n \sim U(n_{min}, n_{max})$ {Sample number of growth steps}
11: $\quad$ // Forward pass
12: $\quad x \leftarrow \mathcal{B}$
13: $\quad$ **for** $t = 1$ **to** $n$ **do**
14: $\quad\quad \Delta x \leftarrow f_\theta(x)$ {Compute update}
15: $\quad\quad x \leftarrow x + \Delta x$ {Apply update}
16: $\quad$ **end for**
17: $\quad x_{rgba} \leftarrow x_{:,:4}$ {Extract RGBA channels}
18: $\quad$ // Compute loss and update
19: $\quad \mathcal{L} \leftarrow \frac{1}{B} \sum_{i=1}^{B} \text{MSE}(x_{rgba}^{(i)}, I)$
20: $\quad \nabla\theta \leftarrow \nabla_\theta \mathcal{L}$
21: $\quad \nabla\theta \leftarrow \frac{\nabla\theta}{||\nabla\theta||+\epsilon}$ {Normalize gradients}
22: $\quad$ Update $\theta$ using optimizer
23: $\quad$ // Update pool
24: $\quad$ losses $\leftarrow \{\text{MSE}(x_{rgba}^{(i)}, I)\}_{i=1}^{B}$
25: $\quad \mathcal{W} \leftarrow$ TopK(losses, $k = \lfloor 0.15B \rfloor$) {Worst 15%}
26: $\quad$ **for** $i \in \mathcal{W}$ **do**
27: $\quad\quad x^{(i)} \leftarrow x_0$ {Replace with initial state}
28: $\quad$ **end for**
29: $\quad$ Update pool with new states
30: **end for**

---

an increasing number of rules tend to exhibit lower KL divergence, suggesting that greater rule diversity enhances the model's ability to approximate the target distribution.

However, this improvement is not strictly linear, and we observe diminishing returns beyond a certain number of rules. This saturation effect suggests that while additional mixture components increase flexibility, excessive complexity does not necessarily translate into significant performance gains. Furthermore, we find that introducing internal stochasticity in the model slightly reduces KL divergence for this task. These findings support the idea that an excessively high number of rules can be detrimental, as performance improvements do not sufficiently compensate the increased computational cost.

## E   Comparison with Approximate Bayesian Computation

We implement a standard Approximate Bayesian Computation (ABC) approach to infer the parameters of our agent-based tissue growth model and compare it with the MNCA results. The parameter space $\Theta$ is the same as the original model in Appendix A, here we report the sampling distribution:

- division rates ($\vec{b} \in \mathbb{R}^5 \sim \text{Gamma}(1, 0.1)$)

- death rates ($\vec{d} \in \mathbb{R}^5 \sim \text{Gamma}(1, 0.01)$)

- survival rates ($\vec{s} \in \mathbb{R}^5 \sim \text{Gamma}(1, 0.1)$)

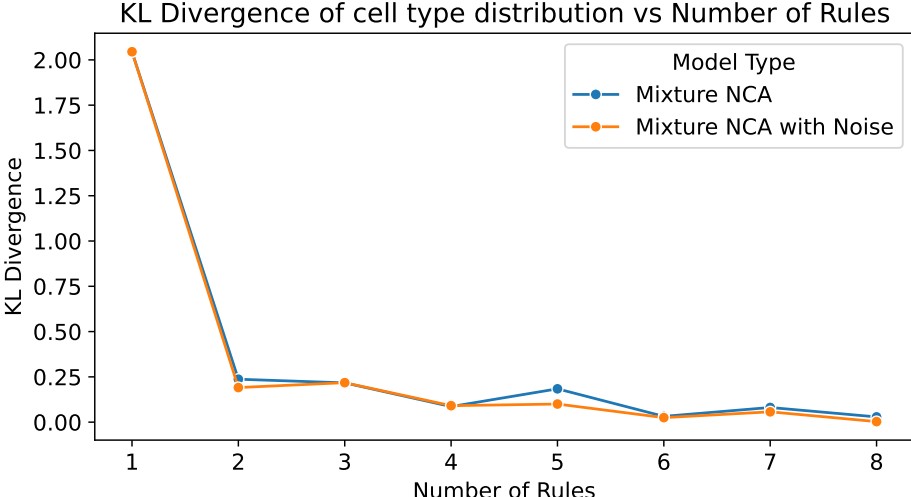

Figure 10: **KL divergence of the cell-type distribution as a function of the number of rules used in the Mixture NCA model.** As the number of rules increases, the divergence between the simulated and target distributions decreases, indicating improved alignment with the expected cell-type dynamics.

- differentiation rates ($D \in \mathbb{R}^{5 \times 5} \sim \mathrm{Gamma}(1, 0.1)$)

- cell-cell interaction strengths ($I \in \mathbb{R}^{5 \times 5} \sim \mathcal{N}(0, 1)$)

---

**Algorithm 4** ABC for Tissue Growth Model

---

**Require:** Observed data $\mathcal{D}$, number of particles $N$, acceptance threshold $\epsilon$, summary statistic type $S$
**Ensure:** Estimated parameters $\theta^*$
 1: Initialize empty sets $\mathcal{A}_\theta, \mathcal{A}_\delta$ for accepted parameters and distances
 2: **for** $i = 1$ to $N$ **do**
 3:     Sample $\theta_i \sim p(\theta)$ from prior distributions
 4:     Simulate tissue growth $x_i \sim f(\cdot | \theta_i)$ for $T$ steps
 5:     Compute summary statistic $s_i = S(x_i)$
 6:     Calculate distance $\delta_i = Dist(s_i, S(\mathcal{D}))$
 7:     **if** $d_i < \epsilon$ **then**
 8:         $\mathcal{A}_\theta \leftarrow \mathcal{A}_\theta \cup \{\theta_i\}$
 9:         $\mathcal{A}_\delta \leftarrow \mathcal{A}_\delta \cup \{\delta_i\}$
10:     **end if**
11: **end for**
12: Compute weights $w_i = 1/\delta_i$, normalized
13: Return $\theta^* = \sum_i w_i \theta_i$ for $\theta_i \in \mathcal{A}_\theta$

---

We trained three different models each with specific summary statistics to compare simulated and observed data:

1. **Cell-type Distribution**: Captures the global proportion of each cell type, including empty spaces. The distance between distributions is computed using the Wasserstein metric $W_1$. This statistic provides a high-level view of tissue composition but does not capture spatial organization.

Table 4: **Comparison of ABC inference on agent-based models of the simulation**

| MODEL | KL-DIV | SIZE-W | BORDER-W |
|---|---|---|---|
| ABM MODEL PROPORTION | **0.152 ±0.002** | **0.241 ±0.010** | **0.054 ±0.006** |
| ABM MODEL NEIGHBORHOOD | 0.857 ±0.010 | 0.489 ±0.025 | 0.284 ±0.010 |
| ABM MODEL CORRELATION | 0.386 ±0.010 | **0.241 ±0.012** | 0.055 ±0.009 |

2. **Neighborhood Composition**: Computes the average composition of 3×3 neighborhoods around each position, including empty spaces. This metric captures local spatial patterns and cell-type clustering, using the Wasserstein distance for comparison.

3. **Cell-type Correlation Matrix**: Quantifies pairwise correlations between spatial distributions of cell types. Each entry $R_{ij}$ represents the Pearson correlation coefficient between the binary masks of types $i$ and $j$, with positive values indicating co-occurrence and negative values suggesting spatial segregation. The distance between correlation matrices is computed using the normalized Frobenius norm $\|R_i - R_{\mathcal{D}}\|_F / \sqrt{2}$.

Parameters are accepted if their distance is below the threshold $\epsilon$, and final estimates are computed as weighted averages of accepted particles, with weights inversely proportional to their distances. The $\epsilon$ in this case have been chosen to accept approximately 10% of the samples (respectively [0.04, 0.4, 0.52]). For the first model, we generated 5000 samples as this is the fastest statistic to compute, while for the others we drew 1000 samples.

We present the results in Table 4, highlighting how performance is significantly influenced by the choice of statistics used for parameter inference. Interestingly, simple cell-type proportions yield the best results, not only in terms of accuracy but also in maintaining consistency across tissue borders and overall size. These results are comparable to our MNCA approach, though in this case, we had to fully specify the model and fine-tune both the statistics and acceptance threshold. It is also crucial to note that our evaluation is based on the KL divergence of cell proportions, which is directly tied to the statistics used for ABC—specifically, the Wasserstein distance between cell-type proportions.

However, the limitations of this approach are well illustrated in Figure 11. A naive selection of statistics may produce models that generate seemingly accurate summary statistics, yet fail to capture the intricate spatial characteristics of real tissue. As a result, while the summary metrics appear realistic, the simulated tissue structure diverges significantly from the actual one. Conversely, models that better preserve spatial coherence tend to exhibit substantial distortions in cell-type proportions.

## F   Why Mixtures Increase Robustness

We speculate that two main effects could explain the robustness of mixture-based update rules relative to a single deterministic rule.

**Ensemble or Averaging Effect.**   Let $\{f_k : \mathcal{S} \to \mathcal{S}\}_{k=1}^K$ be a family of update functions, each with Lipschitz constant $L_k$. Define a *mixture* update $F$ via

$$F(s) \;=\; \sum_{k=1}^K \pi_k(s)\, f_k(s) \quad \text{where} \quad \sum_{k=1}^K \pi_k(s) \;=\; 1. \tag{15}$$

Here $\{\pi_k(s)\}_{k=1}^K \geq 0$ are mixture weights that may depend on the state $s$. If each $f_k$ is $L_k$-Lipschitz, i.e.,

$$\|f_k(s) - f_k(t)\| \;\leq\; L_k \, \|s - t\|, \tag{16}$$

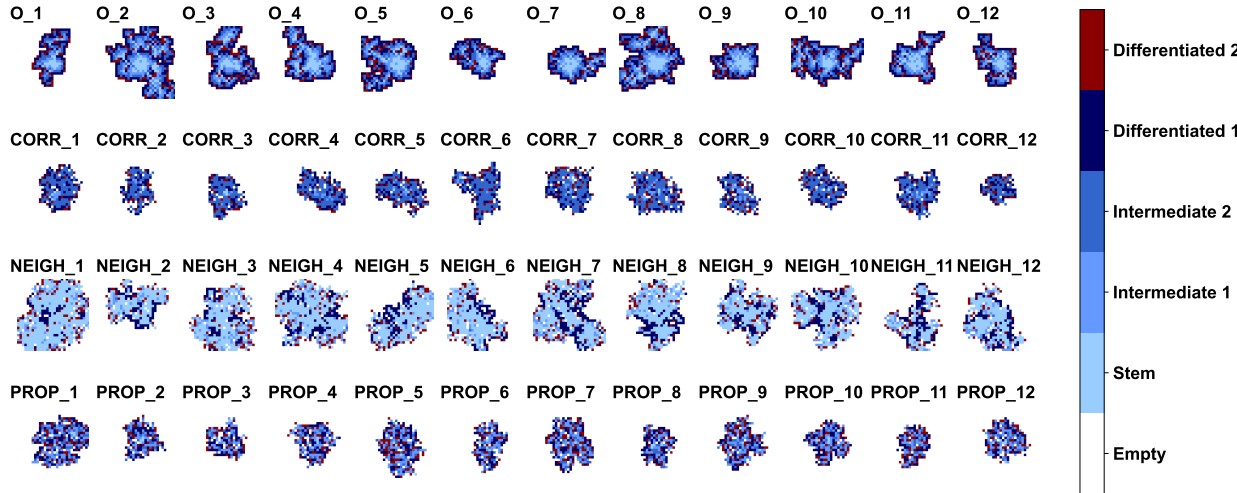

Figure 11: **Simulated tissues by different agent-based models trained with ABC on the simulated dataset.** Top row is 10 tissues from the training data. The second row are tissues generated by the ABC schema with type correlation as summary statistics. The third row are tissues generated by the ABC schema with neighborhood composition as summary statistics. The last row has cell-type distribution as summary statistics.

then $F$ itself is Lipschitz with a constant bounded by $\sum_{k=1}^{K} \pi_k(s) L_k$ (under mild regularity conditions on $\pi_k$). Intuitively, since no single $f_k$ fully dictates the update, large "jumps" from one sub-model are moderated by others. This yields an error-averaging effect, often translating into smoother global dynamics and improved tolerance to perturbations.

To validate this claim we exploited the fact that in our network we can bound from above the Lipschitz constant using the product of the spectral norm of the linear layers Miyato et al. (2018). Figure F shows that although some individual rules exhibit higher bounds, the average bound is lower in the mixture models. This suggests that the mixture can exploit different rules to balance updates of varying magnitudes. It also potentially explains the overshoot right after a perturbation, due to the model uncertainty to which set of rules to apply.

**Randomness and Escaping Attractors.** A purely deterministic update $f(s)$ can exhibit limit cycles or fixed-point attractors, leading to "lock-in" where the system remains trapped. In contrast, a stochastic mixture approach samples an index $k \sim \pi(\cdot \mid s)$ at each iteration—where $\pi(\cdot \mid s)$ is a probability distribution over distinct update functions $\{f_1, \ldots, f_K\}$—and applies $s \mapsto f_k(s)$. This defines a Markov chain on the state space $\mathcal{S}$ with transition kernel

$$P(s, s') = \sum_{k=1}^{K} \pi_k(s) \, \mathbf{1}\big\{ s' = f_k(s) \big\}. \tag{17}$$

Because of the random choice of update rule, the system can sometimes "jump" out of local cycles or fixed points that would be stable under any single deterministic rule.

If the chain is ergodic we are sure that we will eventually get out of a specific state. However, even if such conditions do not strictly hold[2], introducing randomness often increases global stability: stochasticity can help the system avoid being frozen in narrow attractors by tempering a purely deterministic update rule.

---

[2]For instance if the final image is a proper absorbing state for each pixel, which is something we can reasonably expect from a system that has reliably learned to reproduce an image

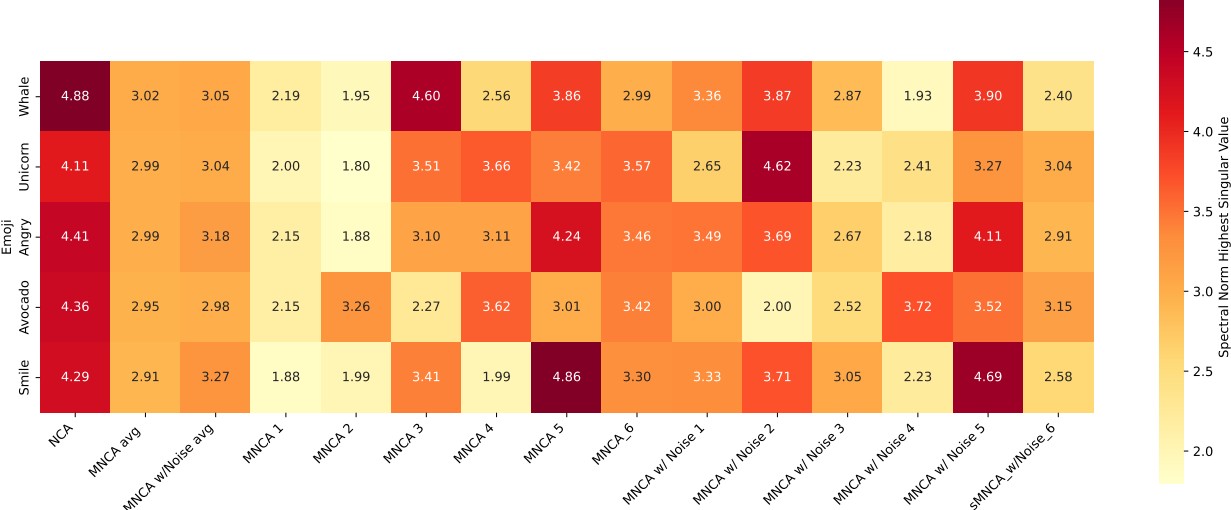

Figure 12: **Heatmap of each model's product of linear layers singular.** In our simple architecture, this product is an upper bound on the Lipschitz constant of the network. Lower values imply tighter Lipschitz bounds, suggesting greater stability under perturbations.

While testing this in the stochastic model is not straightforward we found several repeated patterns when looking at the reconstruction pattern after perturbation of the deterministic NCA. We show some in Figure 13. In particular, we see how the "cut in half" whale seems to arise quite frequently, together with a state where the eyes multiply over the body, and for the avocado emoji, a state where the fruit is smaller and without the seed.

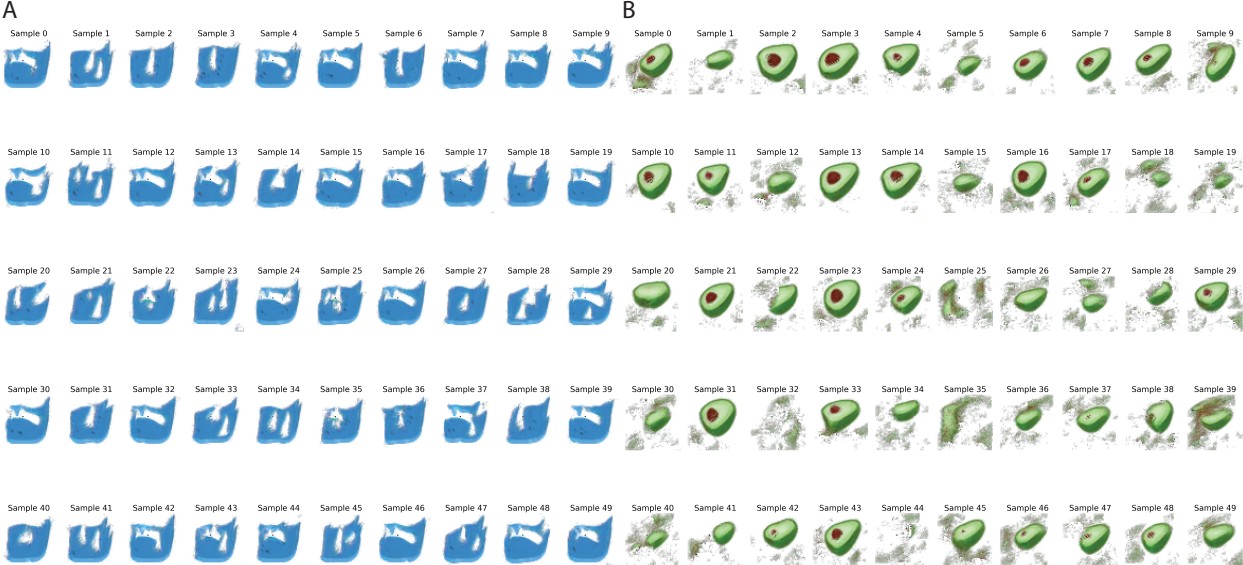

Figure 13: **Final reconstructed image after perturbation for the deterministic NCA.** We took as an example the perturbation with Gaussian injected noise in 25% of the pixels. This is the final image after 100 steps of recovery. A is the whale emoji (1F433) and B the avocado emoji (1F951)

## G  Further evidence on Image Morphogenesis

In this section, we first illustrate in Figure 16 that all three models are correctly trained and capable of generating high-quality realizations of each emoji analyzed in Section 4.3. This validates our interest in assessing their responses to perturbations. To further support our claim that MNCA exhibits superior perturbation robustness compared to NCA, we replicated the perturbation experiment from the main text using nine images from the CIFAR-10 dataset—one per class. Unlike the emoji dataset, CIFAR-10 features more complex images with backgrounds, but at a lower resolution. We used the original $32 \times 32$ RGB images, maintaining all previous training parameters except for the hidden dimension size of the update network, which was reduced to 64 neurons to match the lower resolution. The alpha channel was set to 1. Our findings align with the results of the emoji experiment: MNCA and MNCA with noise consistently outperformed NCA across all perturbations (Table 5).

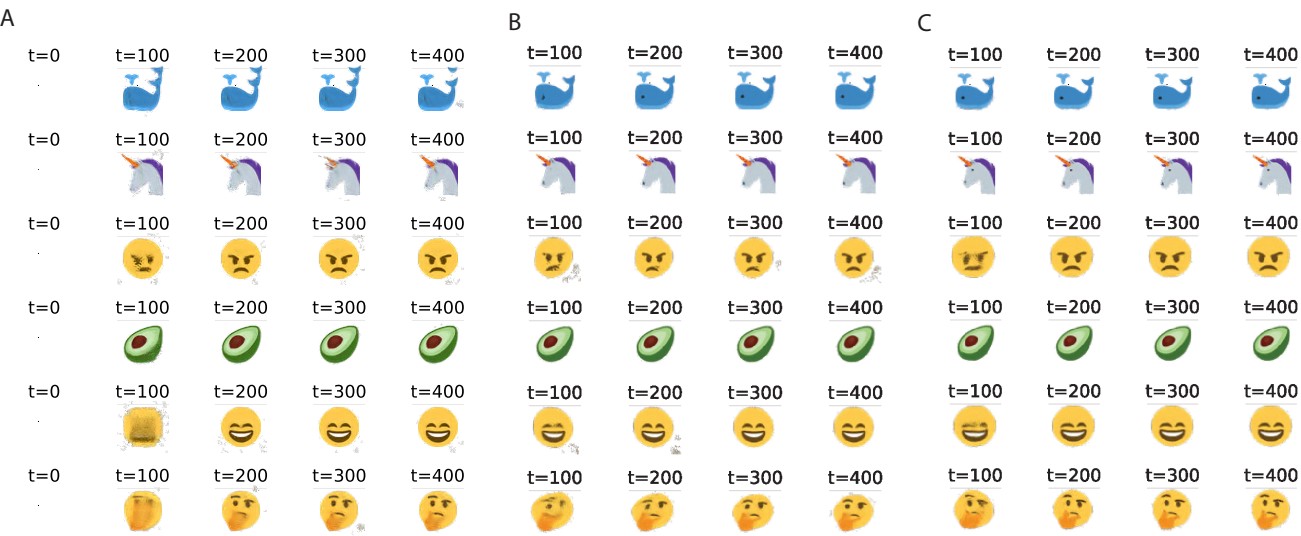

Figure 14: **Emoji morphogenesis for each NCA class.** We show how all three 3 NCA classes converge to a visually good final image from the fixed seed at t=0. A, B, and C are respectively the deterministic NCA, the MNCA, and the MNCA with intrinsic noise.

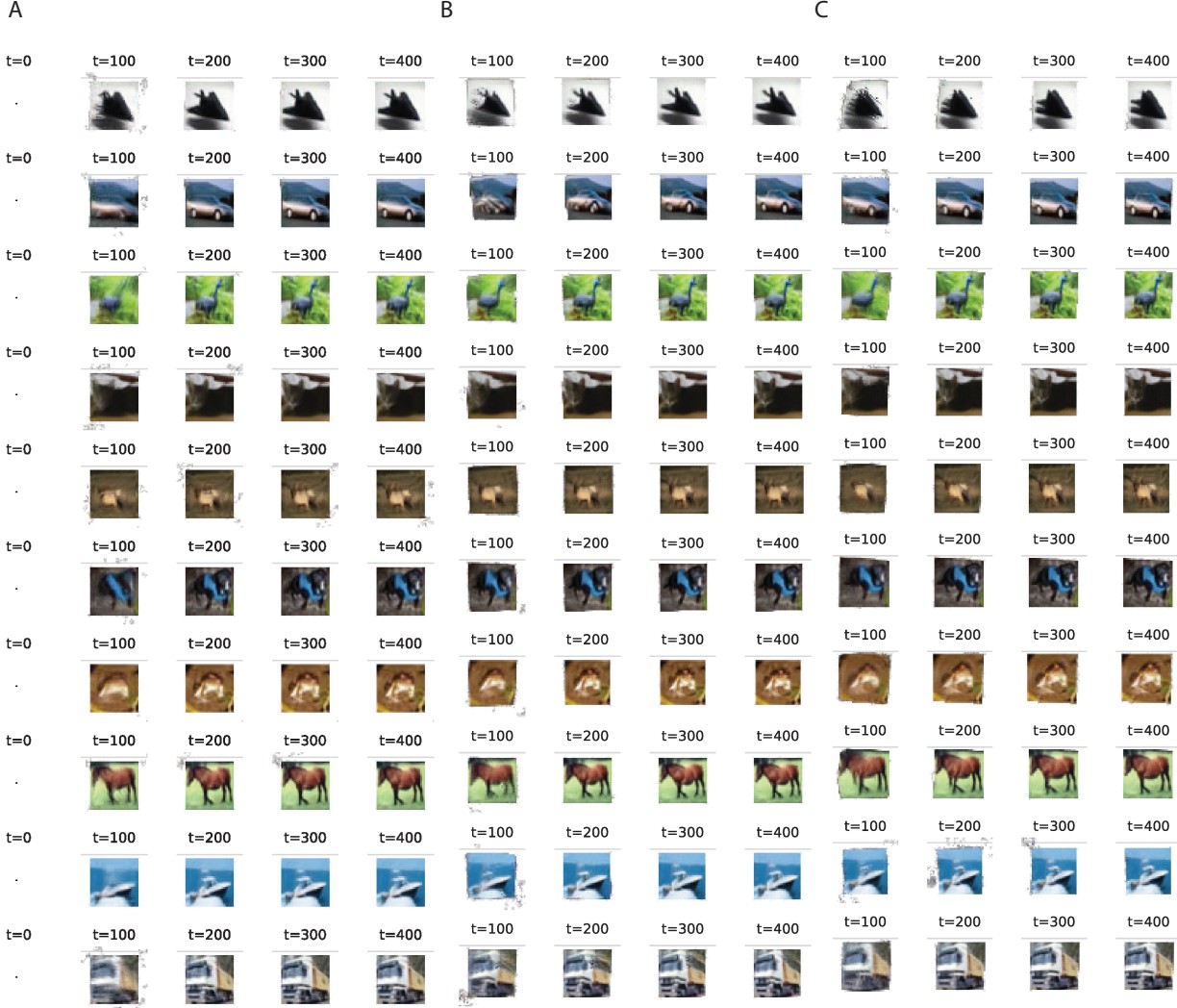

Figure 15: **CIFAR-10 morphogenesis for each NCA class.** We show how all three 3 NCA classes converge to a visually good final image from the fixed seed at t=0. A, B, and C are respectively the deterministic NCA, the MNC,A and the MNCA with intrinsic noise.

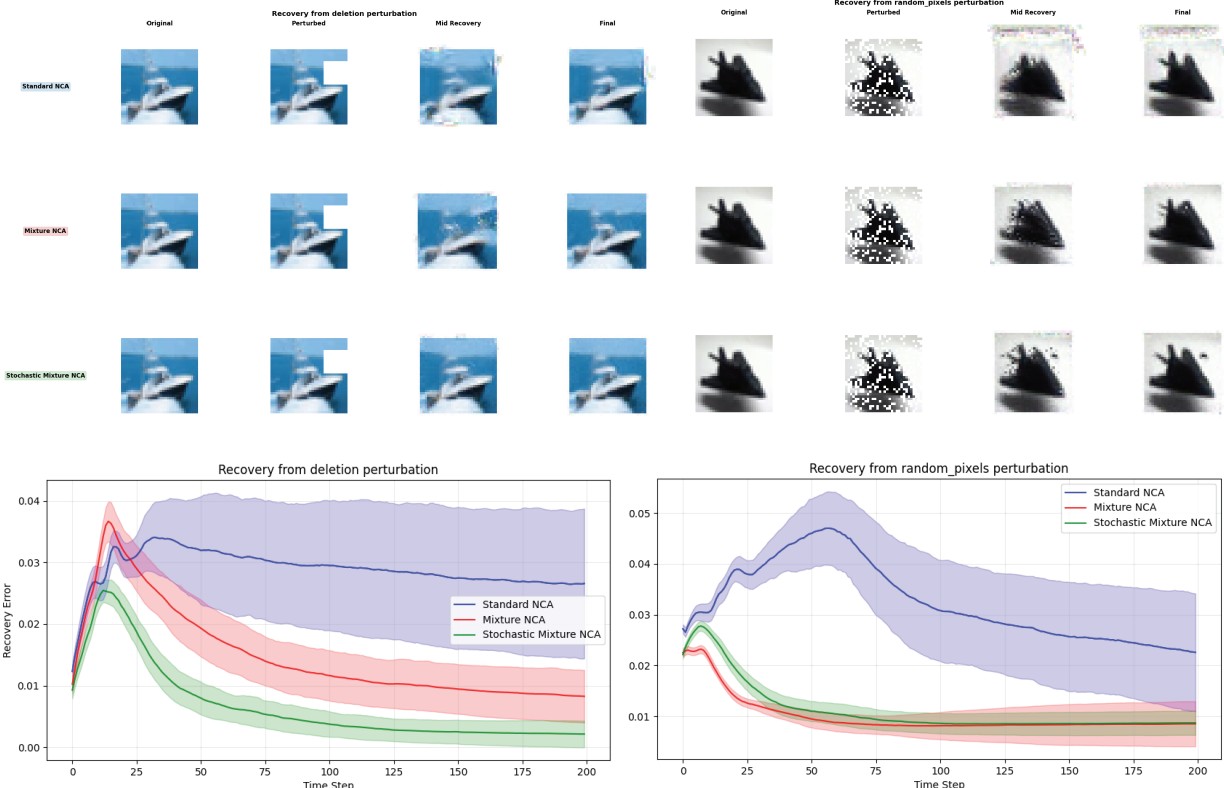

Figure 16: **Example of perturbation and recovery on CIFAR-10 dataset** The top half of the plot shows an example of 2 perturbation types. The columns are respectively, the original image, the perturbed image, the image after 50 steps of recovery, and the final recovered image after 100 steps. Each row shows a separate model. The second half of the plot shows the same image and perturbation above the MSE error behavior from the perturbation time to the final recovery step. The plot shows two cases, respectively, one in which the NCA seems to diverge but the MNCA does not, and the other case where all three models seem to go back to the original image but with different efficiency. Confidence intervals (95%) were computed across 50 different perturbation of the same kind

Table 5: **Performance of MNCA across different perturbations on images from the CIFAR-10 dataset.** Here, the metric is the Mean Squared Error (MSE) after 100 steps of recovery and on a batch of 50 different perturbations of each kind. MNCA w/ N stands for MNCA with Noise, Del for Deletion and Rem for Pixel Removal.

| Model | Del 5x5px | Del 10x10px | Noise 10% | Noise 25% | Rem 100px | Rem 500px |
|---|---|---|---|---|---|---|
| **airplane** | | | | | | |
| NCA | 0.0213 ±0.0083 | 0.0226 ±0.0117 | 0.0213 ±0.0095 | 0.0198 ±0.0087 | 0.0149 ±0.0098 | 0.0222 ±0.0106 |
| GCA | 0.0090 ±0.0055 | 0.0089 ±0.0051 | 0.0103 ±0.0061 | 0.0092 ±0.0058 | 0.0090 ±0.0046 | **0.0108 ±0.0051** |
| MNCA | **0.0012 ±0.0012** | **0.0085 ±0.0045** | **0.0033 ±0.0032** | **0.0010 ±0.0008** | 0.0132 ±0.0024 | 6.5645 ±22.8378 |
| MNCA w/ N | 0.0014 ±0.0020 | 0.0087 ±0.0024 | 0.0040 ±0.0041 | **0.0010 ±0.0011** | **0.0081 ±0.0044** | 0.0110 ±0.0047 |
| **automobile** | | | | | | |
| NCA | 0.0095 ±0.0045 | 0.0155 ±0.0053 | 0.0091 ±0.0045 | 0.0083 ±0.0036 | 0.0139 ±0.0054 | 0.0210 ±0.0055 |
| GCA | 0.0089 ±0.0084 | 0.0093 ±0.0110 | 0.0137 ±0.0132 | 0.0095 ±0.0098 | 0.0095 ±0.0087 | 0.0142 ±0.0110 |
| MNCA | **0.0027 ±0.0016** | 0.0058 ±0.0022 | 0.0037 ±0.0021 | **0.0024 ±0.0015** | 0.0092 ±0.0022 | 0.0178 ±0.0022 |
| MNCA w/ N | 0.0038 ±0.0024 | **0.0045 ±0.0029** | **0.0034 ±0.0022** | 0.0038 ±0.0025 | **0.0047 ±0.0035** | **0.0099 ±0.0043** |
| **bird** | | | | | | |
| NCA | **0.0053 ±0.0037** | **0.0053 ±0.0033** | **0.0073 ±0.0049** | 0.0073 ±0.0048 | **0.0065 ±0.0043** | **0.0091 ±0.0051** |
| GCA | 0.0335 ±0.0083 | 0.0370 ±0.0110 | 0.0327 ±0.0110 | 0.0351 ±0.0104 | 0.0321 ±0.0121 | 0.0333 ±0.0112 |
| MNCA | 0.0066 ±0.0030 | 0.0071 ±0.0029 | **0.0073 ±0.0033** | **0.0052 ±0.0030** | 0.0071 ±0.0033 | 0.0094 ±0.0028 |
| MNCA w/ N | 0.0159 ±0.0040 | 0.0162 ±0.0044 | 0.0163 ±0.0047 | 0.0157 ±0.0052 | 0.0181 ±0.0043 | 0.0183 ±0.0047 |
| **cat** | | | | | | |
| NCA | 0.0135 ±0.0057 | 0.0124 ±0.0062 | 0.0146 ±0.0046 | 0.0134 ±0.0052 | 0.0150 ±0.0068 | 0.0147 ±0.0061 |
| GCA | 0.0059 ±0.0064 | 0.0045 ±0.0044 | 0.0066 ±0.0060 | 0.0065 ±0.0065 | 0.0101 ±0.0094 | 0.0108 ±0.0074 |
| MNCA | **0.0010 ±0.0016** | **0.0010 ±0.0014** | **0.0009 ±0.0012** | **0.0007 ±0.0010** | **0.0006 ±0.0008** | 0.0082 ±0.0045 |
| MNCA w/ N | 0.0016 ±0.0018 | 0.0061 ±0.0028 | 0.0030 ±0.0031 | 0.0015 ±0.0020 | 0.0009 ±0.0012 | **0.0061 ±0.0027** |
| **deer** | | | | | | |
| NCA | 0.0082 ±0.0029 | 0.0103 ±0.0029 | 0.0089 ±0.0046 | 0.0078 ±0.0029 | 0.0126 ±0.0030 | 0.0164 ±0.0028 |
| GCA | 0.0036 ±0.0038 | **0.0029 ±0.0029** | 0.0044 ±0.0041 | 0.0043 ±0.0040 | 0.0118 ±0.0034 | 0.0121 ±0.0025 |
| MNCA | 0.0026 ±0.0015 | 0.0032 ±0.0012 | 0.0026 ±0.0013 | 0.0025 ±0.0014 | 0.0076 ±0.0018 | 0.0096 ±0.0016 |
| MNCA w/ N | **0.0005 ±0.0002** | **0.0029 ±0.0012** | **0.0010 ±0.0009** | **0.0005 ±0.0002** | **0.0014 ±0.0012** | **0.0016 ±0.0014** |
| **dog** | | | | | | |
| NCA | 0.0059 ±0.0092 | 0.0044 ±0.0040 | 0.0070 ±0.0082 | 0.0097 ±0.0112 | 0.0068 ±0.0067 | 0.0064 ±0.0071 |
| GCA | **0.0032 ±0.0042** | **0.0028 ±0.0047** | **0.0034 ±0.0037** | **0.0024 ±0.0034** | **0.0033 ±0.0052** | **0.0048 ±0.0056** |
| MNCA | 0.0048 ±0.0021 | 0.0064 ±0.0022 | 0.0065 ±0.0024 | 0.0047 ±0.0016 | 0.0053 ±0.0020 | 0.0065 ±0.0015 |
| MNCA w/ N | 0.0040 ±0.0021 | 0.0045 ±0.0015 | 0.0045 ±0.0025 | 0.0037 ±0.0019 | 0.0079 ±0.0020 | 0.0133 ±0.0022 |
| **frog** | | | | | | |
| NCA | 0.0102 ±0.0076 | 0.0106 ±0.0062 | 0.0117 ±0.0085 | 0.0117 ±0.0058 | 0.0088 ±0.0053 | 0.0157 ±0.0082 |
| GCA | 0.0262 ±0.0081 | 0.0270 ±0.0089 | 0.0301 ±0.0102 | 0.0247 ±0.0082 | 0.0282 ±0.0087 | 0.0266 ±0.0090 |
| MNCA | **0.0013 ±0.0014** | **0.0030 ±0.0024** | **0.0020 ±0.0018** | **0.0013 ±0.0015** | **0.0025 ±0.0019** | **0.0078 ±0.0027** |
| MNCA w/ N | 0.0027 ±0.0016 | 0.0042 ±0.0021 | 0.0034 ±0.0014 | 0.0029 ±0.0012 | 0.0092 ±0.0022 | 0.0148 ±0.0022 |
| **horse** | | | | | | |
| NCA | 0.0104 ±0.0054 | 0.0158 ±0.0060 | 0.0096 ±0.0049 | 0.0100 ±0.0061 | 0.0167 ±0.0044 | 0.0207 ±0.0056 |
| GCA | 0.0269 ±0.0134 | 0.0282 ±0.0109 | 0.0268 ±0.0112 | 0.0281 ±0.0126 | 0.0251 ±0.0094 | 0.0301 ±0.0095 |
| MNCA | 0.0042 ±0.0027 | **0.0042 ±0.0026** | **0.0046 ±0.0028** | **0.0044 ±0.0025** | 0.0041 ±0.0029 | 0.0100 ±0.0041 |
| MNCA w/ N | **0.0028 ±0.0030** | 0.0044 ±0.0041 | 0.0064 ±0.0052 | 0.0047 ±0.0039 | **0.0031 ±0.0034** | **0.0063 ±0.0043** |
| **ship** | | | | | | |
| NCA | 0.0274 ±0.0103 | 0.0206 ±0.0108 | 0.0266 ±0.0123 | 0.0277 ±0.0094 | 0.0268 ±0.0109 | 0.0290 ±0.0127 |
| GCA | 0.0063 ±0.0054 | 0.0148 ±0.0084 | 0.0123 ±0.0205 | 0.0036 ±0.0040 | 0.0061 ±0.0048 | 0.0193 ±0.0067 |
| MNCA | 0.0074 ±0.0034 | 0.0083 ±0.0038 | 0.0083 ±0.0043 | 0.0066 ±0.0038 | 0.0069 ±0.0044 | 0.0076 ±0.0039 |
| MNCA w/ N | **0.0013 ±0.0014** | **0.0018 ±0.0017** | **0.0021 ±0.0022** | **0.0013 ±0.0016** | **0.0056 ±0.0031** | **0.0060 ±0.0038** |
| **truck** | | | | | | |
| NCA | 0.0327 ±0.0117 | 0.0352 ±0.0090 | 0.0326 ±0.0105 | 0.0309 ±0.0119 | 0.0329 ±0.0100 | 0.0351 ±0.0084 |
| GCA | 0.0158 ±0.0053 | 0.0126 ±0.0049 | 0.0147 ±0.0053 | 0.0134 ±0.0049 | 0.0115 ±0.0054 | 0.0176 ±0.0061 |
| MNCA | 0.0041 ±0.0020 | 0.0098 ±0.0032 | 0.0063 ±0.0034 | 0.0036 ±0.0022 | **0.0034 ±0.0022** | **0.0059 ±0.0033** |
| MNCA w/ N | **0.0027 ±0.0021** | **0.0071 ±0.0038** | **0.0061 ±0.0038** | **0.0025 ±0.0018** | 0.0100 ±0.0025 | 0.0139 ±0.0029 |

