# OpenReview forum: "Mixtures of Neural Cellular Automata: A Stochastic Frame- work for Growth Modelling and Self-Organization"
_TMLR — Rejected by TMLR_

### Review · Reviewer_gYL6 · 2025-06-29

**Summary Of Contributions:**

This paper proposes a method called Mixture of Neural Cellular Automata (MNCA), which combines neural cellular automata with a mixture model framework to capture diverse local behaviors and reproduce stochastic dynamics observed in biological systems. The model incorporates multiple rule networks and uses a probabilistic mechanism to select among them at each spatial location. The approach is demonstrated across several tasks and shows improved performance over baseline methods. While the core idea is interesting and well-motivated, there are several critical concerns that need to be addressed before the paper can be recommended for publication.

**Audience:**

Yes

**Broader Impact Concerns:**

No.

**Claims And Evidence:**

No

**Requested Changes:**

**Questions & Suggestions**

***Major***
1) The authors compared their proposed method, MNCA, with the existing two methods, NCA and GCA, and showed its improved performance. However, I am not sure whether these comparisons with baseline methods are fair or not. The MNCA architecture explicitly incorporates multiple rule networks and a selection mechanism,  allowing it to express a diverse range of local behaviors.  In contrast, the baseline methods are inherently limited to a single, shared update rule and lack any mechanism to capture such heterogeneity.
To ensure a more balanced comparison, I suggest either (i) including additional baselines that can represent multiple local behaviors [Herenandez et al. (2021) or others], or (ii) controlling for total parameter count or representational capacity across models. Without such adjustments, it remains unclear whether MNCA's improvements are due to its architectural novelty or simply increased expressivity.

2) The authors claim in Fig. 3 that different rule networks learn to specialize to different cell types (e.g., Rule 0 for STEM, Rule 1 for INTERMEDIATE_1, etc.). However, from visual inspection, this mapping does not appear to be clean or exclusive. For instance, “Rule 1” appears to include many INTERMEDIATE_2 cells, and “Rule 2” does not seem clearly aligned with any specific cell type. Moreover, since the full cell type is explicitly encoded in the state vector $s_i^t$ (via one-hot encoding), each rule network already has direct access to this information. This undermines the assumption that rules must specialize, since each rule could, in principle, implement multiple update behaviors conditioned on the cell type. There is no architectural constraint or inductive bias that forces one rule per cell type. Supporting this concern, Fig. 10 shows that the KL divergence drops sharply at $K=2$, and barely decreases beyond that. This suggests that two-rule networks may be sufficient to capture the cell dynamics. Lastly, near the end of page 7, the authors state: “When then looking at the rules assignment for the MNCAs...” and make interpretive claims about emergent rule specialization. However, no quantitative evidence is provided to support this assertion such as alignment metrics between rules and cell types. This issue also applies to the explanation involving Fig. 6(C). I suggest either providing supporting analysis or softening the conclusion regarding rule-type correspondence

3) I don’t understand the necessity of stochasticity in this paper. While I agree with the authors’ claim that “stochasticity plays a crucial role in many biological processes”, I am not fully convinced of its necessity within the inference framework proposed in this paper. The presence of intrinsic biological noise is certainly important for understanding the variability and unpredictability in real biological phenomena. However, when the goal is to infer or reconstruct deterministic interaction rules that govern the system’s dynamics, intrinsic stochasticity generally hinders the inference of underlying dynamics.
Furthermore, I question whether the introduction of the Gaussian latent variable $x$ truly reflects biological intrinsic noise. Rather, it appears more akin to a standard noise injection technique, commonly used in deep learning to improve robustness or explore function space more broadly. Alternatively, one could view this mechanism as a form of stochastic sampling (e.g., rejection sampling), designed to help the system escape local minima or capture rare transitions — again, a strategy that serves computational objectives rather than biological fidelity.

4) I am confused about the meaning of “Cat” in Eq. (4a), (5a), (6), and (7). As far as I understand, “Cat” denotes the Categorical distribution to generate the categorical random variable $z$ in Eq. (4a) and (5), while it seems to be used differently in Eq. (6) and (7) (perhaps, concatenation). If so, it should be modified. In addition, I believe that Eq. (7) should be corrected by including the model selection operation with the $z$ variable as Eq. (5c) to prevent misleading.

5) In Sec. 4.4, the evaluation appears to be entirely qualitative, based on visual inspection of the resulting dynamics. Could the authors provide a quantitative assessment of performance?


***Minor***
1) The statement that “stochasticity plays a crucial role in many biological processes” is not supported by citations. While the claim may seem self-evident, adding references to relevant biological studies would help ground the paper in prior work and strengthen the motivation for incorporating stochasticity.
2) Page 5: “We then we studied…” → remove the repeated “we.”
3) Section 4.1 refers to $x_t$ as input to $\pi$, but $s_t$ seems correct. Please clarify.
4) Figs. 4 and 6 have font sizes that are too small to read clearly.
5) On page 7, result interpretations come before metric definitions — consider reordering.
6) In Table 2, “GNCA” should be corrected to “GCA.”
6) The first two columns in the top of Fig. 4 seem repetitive — consider simplifying.
7) As described in Eqs. (4a) and (4b), ${z}$ is introduced as a random variable sampled from a categorical distribution. However, during training, it is implemented as a soft one-hot vector via the Gumbel-Softmax trick, meaning that each component $z_k \in [0, 1]$ and $\sum_k z_k = 1$. Please clarify.

**Strengths And Weaknesses:**

**Strengths**
1) The paper is well-written and clearly situates the proposed approach within the broader context of grid-based modeling for biological systems, particularly as an extension of cellular automata frameworks.
2) The core idea of integrating mixture models with Neural Cellular Automata (MNCA) is conceptually interesting and well-motivated. It represents a natural extension to increase modeling flexibility and heterogeneity.
3) The proposed MNCA model is evaluated across multiple representative domains (synthetic tissue, emoji morphogenesis, microscopy) and shows improved performance over baseline methods.
4) The application of MNCA to the BBBC031v1 microscopy dataset demonstrates the potential utility of the method in real biological image analysis, beyond synthetic tasks.




**Weaknesses**

While the paper presents a novel and well-executed framework, it does not explicitly discuss the limitations of the proposed approach. Below are several key concerns that I believe should be acknowledged in the main text, especially if not addressed during revision:

1) The MNCA framework is significantly more computationally expensive than baseline methods, as it employs multiple neural networks (one per rule) for cell updates.
2) The method assumes that the cell type is explicitly known at each spatial location and encodes this information as a one-hot vector in the cell state. While this may be feasible in synthetic experiments, it is an unrealistic assumption in most real-world biological systems, where cell types must often be inferred rather than observed directly.
3) The motivation for injecting stochasticity (via latent Gaussian vectors and stochastic rule selection) is not fully convincing. Although the authors refer to biological processes as inherently stochastic, it is unclear whether this modeling choice truly reflects intrinsic biological randomness or instead serves as a generic regularization/sampling technique.

---

### Review · Reviewer_W7QD · 2025-08-17

**Summary Of Contributions:**

This paper introduces the mixture of neural cellular automata (MNCA), which extends standard neural cellular automata by introducing a stochastic mixture of multiple local update rules. While deterministic nature of existing NCA methods limits the applicability to real-world biological systems, stochasticity of MNCA enables modeling of the complex behaviors observed in such systems. The efficacy of the proposed approach is validated through several experiments including synthetic tissue growth modeling, image morphogenesis, and microscopy image segmentation.

**Audience:**

Yes

**Broader Impact Concerns:**

This paper explicitly states that the proposed method is applicable to real-world biological systems. From this point of view, it would be desirable for this paper to include at least a minimal broader impact statement similar to those found in other CA studies.

**Claims And Evidence:**

No

**Requested Changes:**

1. Include citations to relevant prior work in the Introduction to clearly position the paper within existing research.
1. Discuss and explain the motivation for using a mixture model rather than other probabilistic methods.
1. Correct grammatical errors throughout the manuscript and perform thorough proofreading.
1. Correct stylistic errors, e.g. tense inconsistencies, lack of parentheses in citation, inconsistent equation reference format, duplicated usage of "Cat" for Categorical and Concatenation.
1. Improve figure captions and in-text figure references so that captions alone convey the main elements and the meaning of colors or labels without relying on the main text.
1. Explicitly define π in the main text as a probability vector and describe its role in the model.
1. Add a concise explanation in the main text of the Gumbel–Softmax trick, why it is needed, and what it enables in this model.
1. Avoid using “one-hot-encoded” as a verb; rephrase using a more standard form (e.g., “encoded each cell type as a one-hot vector”).
1. Add clarifying remarks for the notation φ_k: S^{|N(i)|+1} → S, explaining the meaning of the input and output.
1. Clarify the notation [5, 15] used in the experiments (e.g., for the number of stem cells). In mathematical writing, [a, b] typically denotes a closed interval, while in programming contexts it may denote a list of two elements. To avoid ambiguity, please explicitly state whether this represents a continuous interval, a discrete range, or a fixed set of values (e.g., “uniformly sampled between 5 and 15” or “a random integer between 5 and 15”).

**Strengths And Weaknesses:**

# Strengths
Combining mixture modeling with NCAs in a stochastic framework is an original and relevant idea for modeling complex systems. The experimental results demonstrate the wide applicability of the proposed method across different domains. Moreover, the approach emphasizes improved robustness to perturbations and enhanced interpretability compared to existing baselines.

# Weaknesses
In my opinion, the central novelty of the paper is the introduction of a mixture formulation into NCAs. However, the positioning of this contribution within the broader CA literature remains unclear. As a reviewer without deep expertise in the CA field, I cannot determine whether mixture-based formulations have already been explored in the context of classical Cellular Automata (CA), beyond NCAs. The paper neither cites relevant prior work on CA mixtures (if such work exists) nor states explicitly that, to the best of the authors’ knowledge, no such work exists. This omission makes it difficult for the reader to assess the true novelty of the contribution. this is the main reason why I so far evaluate the "Claims And Evidence" as No.

In addition, while the empirical results are convincing, the paper does not provide sufficient justification for why stochasticity should be introduced specifically through mixtures, rather than through other stochastic mechanisms (e.g., probabilistic rule updates, noise injection, or something). A clearer argument about the necessity and advantages of mixture modeling would strengthen the paper’s contribution.

While the technical content of the paper is interesting, the presentation quality requires significant improvement. There are numerous grammatical issues, inconsistent equation notations, missing parentheses in citations, and non-standard usage of abbreviations. In addition, figure captions are not sufficiently self-contained and often require the reader to consult the main text to interpret them. These problems reduce readability and accessibility. Considering TMLR’s evaluation criteria, improvements in presentation should be regarded as mandatory.

---

### Review · Reviewer_mAYe · 2025-09-01

**Summary Of Contributions:**

The paper proposes Mixtures of Neural Cellular Automata (MNCA), which extends Neural Cellular Automata (NCA) in two ways: (1) it comprises $K$ distinct NCAs that are combined using a probabilistic rule assignment, and (2) it has Gaussian noise as inputs to model stochasticity in biological systems. The authors demonstrate that MNCAs accurately model biological growth patterns as well as offer increased robustness to perturbations in image morphogenesis.

**Audience:**

Yes

**Broader Impact Concerns:**

There is no negative ethical implications of the work as far as I can see.

**Claims And Evidence:**

Yes

**Requested Changes:**

- A common and easy way to make a model stochastic is by considering dropout during inference, or considering deep ensembles. In particular, the latter is usually found to work quite well on various deep learning systems. Have the authors considered these alternatives to the way that they're making the model stochastic? It is also worthwhile trying these simple methods on the NCA baseline as well.
- Similarly, it might be interesting to consider a mixture-type model for GCA as an alternative. Here, the stochasticity will be modelled by means of the mean and variance of Gaussian rather than as noisy inputs. The authors found that "GCA seem not to have a major advantage over the vanilla NCA in terms of robustness", which may also be due to the limited expressivity of Gaussians. Considering a mixture approach in this framework could be another way to simultaneously increase flexibility and make the model probabilistic.

__Minor comments:__
- The acronym GNCA is not introduced. I presume this is just Gaussian Cellular Automata with a learnable mean and variance, as stated at the end of Section 2.3.
- The bottom plots in Figure 4 can be improved by using a bigger label size overall for better visibility.
- There are several misprints found throughout the text:
    - Third paragraph of Section 4: "We then we studied the behaviour..." -> "We then studied the behaviour..."
    - Footnote 1: "There is, however, a positive contribution fixed contribution..."
    - End of page 8 to beginning of page 9: "It is interesting to note how none MNCAs the two different flavors of MNCA..."
    - Caption of Figure 8: "The top row is are trajectory from the training dataset"
    - Caption of Figure 15: "MNC,A" -> "MNCA"

**Strengths And Weaknesses:**

__Strengths:__
- The paper is generally well-written, especially towards the beginning, and realtively easy to follow.
- Experimental results show that MNCA (with or without noise) is capable of learning the generative dynamics of biological growth patterns, with much lower probabilistic scores, and consistently demonstrates significant robustness to image perturbations for morphogenesis.

__Weaknesses:__
- It appears all models are trained in the same way by minimizing the MSE with the ground truth. However, I am not sure if that is entirely reasonable for the stochastic model, which seems better suited for training using probabilistic scores such as log likelihood or CRPS. Or, if the underlying biological system is stochastic, then a score measuring the distance between the true underlying measure and the measure of the stochastic MNCA, e.g., using the KL divergence, Wasserstein metric, or MMD, might be more appropriate. The same can be said about the GNCA baseline.
- It seems like the performance of MNCA with vs. without noise can vary quite significantly for some data. For example, in the results in Table 3, MNCA without noise sometimes achieves much better results than with noise (e.g., happy emoji with noise 25\%, all perturbations with  unicorn emoji, etc), while on some other data the opposite happens (e.g. happy emoji with pixel deletion). This inconsistency can make it hard to determine which model to use in which situations.
- The writing seems to get rushed towards the end, with several misprints found (see requested changes below).

---

### Decision · Action_Editor_LUN4 · 2025-10-01

**Recommendation:** Reject

**Additional Comments:**

The paper proposes a method to make neural cellular automata more probabilistic via a mixture distribution and added Gaussian noise. It is shown how empirically how this helps in learning a range of dynamical systems.

All the reviewers had similar serious concerns. Unfortunately, the authors did not respond to these during the rebuttal period. Therefore rejection is recommended.

**Audience:**

Yes

**Audience Explanation:**

Yes, the paper proposes a model learning dynamics.

**Claims And Evidence:**

No

**Claims Explanation:**

The authors experimentally validate their proposed method. However, a number of central questions posed by the reviewers are left unanswered.